# Synaptic vesicle proteins are selectively delivered to axons in mammalian neurons

Emma T Watson[1,2], Michaela M Pauers[1,2†], Michael J Seibert[1,2], Jason D Vevea[1,2‡], Edwin R Chapman[1,2*]

[1]Department of Neuroscience, University of Wisconsin-Madison, Madison, United States; [2]Howard Hughes Medical Institute, Madison, United States

**Abstract** Neurotransmitter-filled synaptic vesicles (SVs) mediate synaptic transmission and are a hallmark specialization in neuronal axons. Yet, how SV proteins are sorted to presynaptic nerve terminals remains the subject of debate. The leading model posits that these proteins are randomly trafficked throughout neurons and are selectively retained in presynaptic boutons. Here, we used the RUSH (retention using selective hooks) system, in conjunction with HaloTag labeling approaches, to study the egress of two distinct transmembrane SV proteins, synaptotagmin 1 and synaptobrevin 2, from the soma of mature cultured rat and mouse neurons. For these studies, the SV reporter constructs were expressed at carefully controlled, very low levels. In sharp contrast to the selective retention model, both proteins selectively and specifically entered axons with minimal entry into dendrites. However, even moderate overexpression resulted in the spillover of SV proteins into dendrites, potentially explaining the origin of previous non-polarized transport models, revealing the limited, saturable nature of the direct axonal trafficking pathway. Moreover, we observed that SV constituents were first delivered to the presynaptic plasma membrane before incorporation into SVs. These experiments reveal a new-found membrane trafficking pathway, for SV proteins, in classically polarized mammalian neurons and provide a glimpse at the first steps of SV biogenesis.

## Editor's evaluation

The authors explored a key question in nerve cell biology, i.e. how these highly polarized cells achieve the specific and differential distribution of proteins and organelles into their axonal and dendritic compartments – the study is an important step forward in this context. By using a very-low-level expression paradigm to express fluorescently tagged reporter proteins in neurons, a method to allow their triggered and 'synchronous' exit from the endoplasmic reticulum (RUSH), and live cell imaging, the authors describe a specific axonal trafficking pathway for the synaptic vesicle proteins Synaptotagmin-1 and Synaptobrevin-2. The corresponding evidence is compelling, and, furthermore, the authors' observation that even slightly excessive expression levels of the fluorescently tagged reporters occlude the specific axonal trafficking so that proteins distribute indiscriminately into axons and dendrites, explains why previous studies often failed to detect specific axonal trafficking of synaptic vesicle proteins. This study will be of interest to cell biologists and neuroscientists alike because (i) it provides a major advance in our understanding of nerve cell development and function, (ii) it demonstrates the usefulness of the RUSH approach in nerve cell biology, and (iii) it stresses the importance of tight control of reporter (over)expression, which is important in many other contexts.

**\*For correspondence:**
chapman@wisc.edu

**Present address:** †Doctoral Program in Neurobiology and Behavior, Department of Neuroscience, Columbia University, Jerome L. Greene Science Center, New York, United States; ‡Department of Developmental Neurobiology, St. Jude Children's Research Hospital, Memphis, United States

**Competing interest:** The authors declare that no competing interests exist.

## Introduction

Neurons present a dramatic example of cell polarization. These highly specialized and asymmetric cells form elaborate axonal and dendritic arbors, with some axons extending great distances (e.g., axons in a blue whale can reach a length of 30 meters; *Smith, 2009*). Within this polarized framework, axons and dendrites are highly adapted to carry out different functions and, consequently, each harbor somewhat distinct molecular constituents. For example, in chemical synapses, dendrites require a steady supply of receptors and proteins that are involved in postsynaptic signaling, whereas axons require the machinery that drives the synaptic vesicle (SV) cycle, including the exocytosis of neurotransmitters. How this molecular and cellular polarity is maintained, specifically in the case of highly extended axons, is an essential question since preserving this extreme polarity underlies neuronal function. Indeed, defects in axonal transport have been implicated in a variety of neurodegenerative diseases (*Hung and Link, 2011*; *Maday et al., 2014*; *May-simera and Liu, 2013*; *Vicario-Orri et al., 2014*).

Many aspects of axonal and dendritic transport are well characterized (*Hirokawa, 1993*; *Maday et al., 2014*; *Roy, 2014*; *Twelvetrees, 2020*). Specifically, several families of motor proteins, which carry transport vesicles along microtubule and actin tracks, have been described (*Hirokawa and Takemura, 2005*; *Kneussel and Wagner, 2013*). Together, motor proteins and the cytoskeleton constitute a transport network that supports the formation and maintenance of synapses (*Waites et al., 2005*; *Ziv and Garner, 2004*). In the case of the axonal transport of SV proteins, anterograde movement is driven by the kinesin motors, KIF1A (*Okada et al., 1995*) and KIF5B (*Nakata and Hirokawa, 2003*; *Song et al., 2009*), and retrograde transport is mediated by dynein (*Fejtova et al., 2009*; *Paschal and Vallee, 1987*; *Schnapp and Reese, 1989*). However, how SV proteins are sorted to presynaptic boutons remains unclear.

Considerable progress has been made concerning the postal system by which proteins are selectively sorted to dendrites. A direct pathway, with proteins traveling directly from the soma to dendrites, is thought to be established early in neuronal development (*Burack et al., 2000*; *Karasmanis et al., 2018*; *Petersen et al., 2014*; *Silverman et al., 2001*). Studying the sorting of axonal cargo, specifically SV proteins in mature mammalian neurons, is more challenging, in part due to the limited flux of materials to neurites after synaptogenesis, a point we return to below. Two membrane trafficking pathways that sort SV proteins to axons have been proposed. The first pathway, often called the selective retention model, is based on the non-polarized delivery of axon-destined cargo to both axons and dendrites. In this pathway, SV proteins that were delivered to axons are retained there, whereas SV proteins that were delivered to the plasma membrane (PM) of dendrites are endocytosed and re-routed back to axons (*Fletcher-Jones et al., 2019*; *Sampo et al., 2003*). The second pathway is a variant of this model. It differs in that axon-destined transport vesicles move through axons and dendrites, but do not fuse with the dendritic PM (*Burack et al., 2000*; *Nabb and Bentley, 2022*). This second model has been termed 'direct transport', despite the observation that transport vesicles carrying axonal proteins often entered dendrites prior to entering axons, to emphasize the lack of fusion of these vesicles in the somatodendritic domain. After decades of research, the widespread conclusion is that SV proteins are trafficked with a low degree of selectivity into both axons and dendrites of mammalian neurons, and are either selectively retained in, or have partially biased transport toward, axons (*Bentley and Banker, 2016*).

In contrast to these low-selectivity transport models, recent studies suggest the existence of a direct and selective transport pathway of SV proteins to axons in *Caenorhabditis elegans* DA9 bipolar neurons and rat and mouse pseudounipolar dorsal root ganglion cells (*Gumy et al., 2017*; *Li et al., 2016b*). However, DA9 bipolar neurons have a simplified microtubule orientation, and pseudounipolar neurons lack an axon initial segment, have a bifurcating axon, and do not have dendrites, so it is unclear whether the transport observed in these models extends to mammalian neurons with conventional morphology and microtubule polarity. One elegant study, using mouse hippocampal neurons, concluded that temperature-sensitive vesicular stomatitis virus glycoprotein (VSV-G tsO45) underwent directed polarized axonal transport (*Nakata and Hirokawa, 2003*). VSV-G tsO45 is a foreign protein in mammalian neurons, and the untagged version of this protein is sorted to dendrites (*Dotti and Simons, 1990*). Still, the selective trafficking of this viral protein to axons suggests the existence of an axon-specific pathway in mammalian neurons with classically polarized axonal and dendritic arbors.

The main objective of the current study is to use new, improved methods to address whether axonal proteins arrive at their destination via non-polarized delivery, or via direct and specific transport to axons. Specifically, we trace the path that two distinct SV proteins take from the soma to axons in mammalian hippocampal neurons. We focused on synaptotagmin (SYT) 1, a Ca$^{2+}$ sensor that regulates rapid neurotransmitter release (*Chapman, 2008*), and synaptobrevin (SYB) 2, a vesicular (v-) SNARE protein, also known as VAMP2, that assembles into *trans*-SNARE complexes to catalyze membrane fusion (*Südhof and Rothman, 2009*). We note that SYB2 has been used in axonal transport studies for decades and was included here to directly address the idea that its polarized distribution is achieved via either of the two non-polarized delivery models outlined above (*Nabb and Bentley, 2022*; *Sampo et al., 2003*). These two SV proteins were also selected because SYT1, a canonical type I transmembrane protein, is co-translationally inserted into the endoplasmic reticulum (ER) (*Perin et al., 1990*; *Shao and Hegde, 2011*), whereas SYB2, a type II tail-anchored protein, is post-translationally inserted into the ER (*Kutay et al., 1995*). Moreover, SYT1 and SYB2 have been shown to be trafficked by different kinesin motors, KIF1A (*Okada et al., 1995*) and KIF5B (*Nakata and Hirokawa, 2003*; *Song et al., 2009*), respectively. The selection of two distinct SV proteins with different topologies, and consequently different biosynthetic pathways, as well as different trafficking motors, enabled us to investigate whether there is a conserved mechanism underlying their polarized distributions.

We reiterate that the low rate of SV protein egress from the soma, once neurons mature and switch from establishing to maintaining polarity, makes it difficult to monitor movement of SV precursors via live-cell imaging, without overexpressing the protein of interest. To overcome these technical challenges, we took advantage of recently developed tools to sequester SV proteins in the ER of mature neurons. We then released them in a synchronized manner, after synaptogenesis, to track their path to synapses after leaving the Golgi (*Boncompain et al., 2012*; *Farías et al., 2016*; *Zahavi et al., 2021*). We combined this system with HaloTag labeling approaches (*Grimm et al., 2015*; *Grimm and Lavis, 2021*; *Los et al., 2008*) to follow the fate of SYT1 and SYB2 as they leave the soma and are ultimately delivered to nerve terminals. In these experiments, careful attention was paid to expression levels since overexpression results in the spillover of SV proteins into inappropriate compartments, thus obscuring polarized transport (*Pennuto et al., 2003*). These experiments uncovered a novel pathway in which transport vesicles, bearing transmembrane SV proteins, are directly and selectively delivered to axons. Moreover, after delivery to axons, we found that these vesicles fuse with the presynaptic PM creating a hub, or reservoir, from which SVs are eventually generated.

## Results

### Using RUSH to study egress of SV proteins from the soma of cultured neurons

We took advantage of the retention using selective hooks (RUSH) system to study the sorting itinerary of newly synthesized SV proteins (*Figure 1A*; *Boncompain et al., 2012*). In the RUSH system, proteins of interest are retained in the ER and released upon addition of the small molecule, biotin. The ability to synchronize protein release from the ER makes it possible to observe their trek to their target destination. In our experiments, retention of the SV proteins, SYT1 and SYB2, was accomplished by appending a streptavidin-binding peptide (SBP) to the intravesicular end of each protein; translocation of the amino-terminus of SYT1 into the ER was aided by the addition of a pre-prolactin leader sequence (*Figure 1A and B*). The tagged proteins bind to the co-expressed streptavidin "hook", which is localized to the ER by a retention signal (Lys·Asp·Glu·Leu; KDEL), thus retaining them. The addition of biotin displaces the SBP and allows natural egress to occur. Each cargo also includes a HaloTag that was used for visualization.

It is known that overexpression can cause SV proteins to mislocalize to other compartments, especially the PM (*Pennuto et al., 2003*). To mitigate this confound, the viruses used to express SYT1 and SYB2 were carefully titrated to achieve a sparse transduction such that only a select few neurons were expressing minimal levels of the tagged protein. To further ensure low levels of expression, cells that had lower than average fluorescence (as compared to other transduced cells on the coverslip) were selected for imaging. Within this low-expression paradigm, live-cell imaging and immunocytochemistry (ICC) confirmed the localization of the SV reporter proteins within the ER prior to biotin-triggered

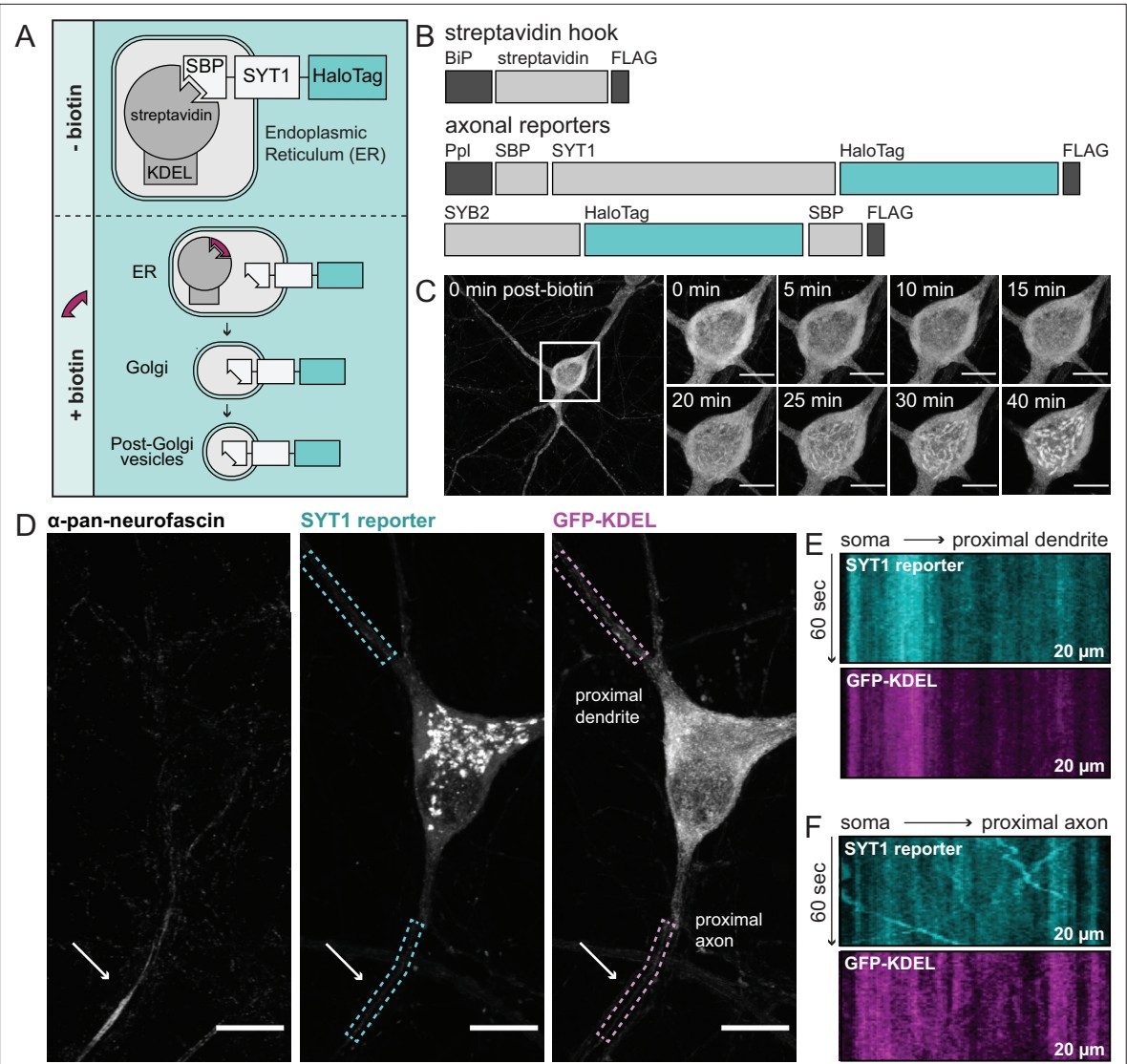

**Figure 1.** Using retention using selective hooks (RUSH) to study egress of synaptic vesicle (SV) proteins from the soma of cultured rat hippocampal neurons. (**A**) A cartoon of RUSH; pre- and post-biotin conditions are shown. (**B**) Schematic of the streptavidin hook, and SYT1 and SYB2 reporter RUSH constructs: BiP, a signal peptide that drives translocation into the ER; FLAG, provides a means to detect each construct; SBP, streptavidin-binding peptide; Ppl, a pre-prolactin leader sequence to translocate the SBP into the endoplasmic reticulum (ER). In all cases the reporter is a HaloTag. (**C**) Representative super-resolution fluorescent live-cell MAX projection images from rat neurons at 15 days in vitro (DIV). Images of SYT1 reporter immediately after biotin addition with enlarged insets to detail the time course of release. Inset scale bar is 10 µm in panels (**C–D**). Since SYT1 and SYB2 behaved similarly, only SYT1 images are shown in panels (**C–F**). (**D**) Image of a neuron, 30 min after biotin addition, expressing the streptavidin hook, SYT1 reporter, and ER-targeted GFP (GFP-KDEL). Live-cell labeling with an anti-pan-neurofascin antibody was used to identify the axon initial segment (AIS; arrow); dendrites were identified by morphology and because they lacked an AIS. SYT1 was labeled with JF549 HaloTag ligand, and kymographs of this reporter, along with GFP-KDEL, were generated from the regions indicated by dashed boxes (20 µm long). Kymographs from a proximal dendrite (**E**) and proximal axon (**F**) are shown.

The online version of this article includes the following figure supplement(s) for figure 1:

**Figure supplement 1.** The SYT1 reporter localizes to the early secretory pathway after biotin addition.

**Figure supplement 2.** The SYB2 reporter is retained in the endoplasmic reticulum prior to biotin addition.

**Figure supplement 3.** SYT1 and SYB2 reporters are targeted to the presynapse.

release, and then with the Golgi and eventual endpoint targeting to synapses following release (*Figure 1C*, *Figure 1—figure supplements 1–3*).

With the temporal control afforded by this assay, we can designate an exact starting position and time of release in the cell, and record trafficking events immediately upon exit from the Golgi, approximately 20–30 min after biotin addition. This is a key point, because without defining a start time and location of release, we cannot know if SV cargoes were trafficked through dendrites on their way to axons. To definitively identify axons, we used an extracellular pan-neurofascin antibody to label the axon initial segment of live neurons (*Figure 1D*). We also expressed an ER-targeted GFP (GFP-KDEL) to determine whether transport vesicles were post-ER organelles, as indicated by the absence of this marker (*Figure 1D*). Upon release of ER-tethered SYT1, we observed little to no transport activity in proximal dendrites (defined as the first 20 μm of the neurite) as shown by the lack of diagonal lines in the kymograph (*Figure 1E*); however, there was noticeable movement of tagged SYT1 as it began to egress from the soma directly into axons in an anterograde direction, as represented in the kymograph by diagonal lines with a negative slope (*Figure 1F*). These initial findings contradict the idea of SV proteins being trafficked with low selectivity into axons and dendrites (*Bentley and Banker, 2016*; *Nabb and Bentley, 2022*; *Sampo et al., 2003*), and thus warranted a deeper examination. We highlight that the majority of anterograde-moving vesicles observed in proximal axons did not contain the co-expressed ER-targeted GFP (*Figure 1F*), confirming that these are post-ER organelles. In contrast, the stationary SYT1 signal in proximal dendrites colocalized with ER-targeted GFP (*Figure 1E*), indicating the observed signal is protein in the ER, rather than post-Golgi transport vesicles. Taken together, the lack of movement in proximal dendrites, and the robust anterograde trafficking in axons, suggest the existence of a selective and specific pathway that sorts SYT1 to presynaptic boutons.

## A direct and selective axonal transport pathway for SYT1 and SYB2

To explore this seemingly novel SV protein trafficking pathway, we expanded our experiments to include a second SV protein, SYB2, and to analyze transport in both proximal and distal regions of both axons and dendrites. Proximal regions were defined as the first 20 μm of a neurite as it emerges from the cell body. Distal regions were defined as a secondary branching of a dendrite or, for axons, a distance of ~150 μm from the soma, which is beyond the axon initial segment (*Figure 2A*). The distal regions were imaged approximately 5 min after the proximal regions to allow time for transport vesicles to make their way farther down neurites (at an average transport rate of 1 μm/s, each transport vesicle has the potential to travel ~300 μm from the soma during this time period). With these selection criteria, transport was quantitatively assessed by generating and analyzing kymographs of post-Golgi vesicles carrying the SV protein of interest (*Figure 2—figure supplement 1*). As was first seen in *Figure 1*, we again observed robust transport of the SYT1 reporter in proximal axons and extended these observations to distal axons; proximal and distal dendrites had little to no detectable trafficking of SYT1-containing transport vesicles (*Figure 2B–D*, *Figure 2—figure supplement 2A*). This trend was also observed for the SYB2 reporter, allowing us to generalize our findings to topologically distinct SV proteins that are transported by different kinesin motors (*Figure 2E–G*, *Figure 2—figure supplement 2B*). Furthermore, all neurites were observed for the exact same duration, so the volume of transport activity in each region can be directly compared, further supporting the idea that there is minimal transport of SV proteins in dendrites. Of the transport vesicles observed, both SYT1 and SYB2 reporters were preferentially transported in the anterograde direction in axons (*Figure 2H and I*). Again, few, if any, puncta were observed in dendrites (*Figure 2J and K*). We note that SYB2 was transported more slowly in the proximal axon, which encompasses the AIS, as compared to the distal axon; SYT1 transport did not slow down in the AIS (*Figure 2—figure supplement 3*; *Song et al., 2009*). These data confirm a general trafficking pathway that—in contrast to previous models (*Nabb and Bentley, 2022*; *Sampo et al., 2003*)—does not include significant flux through dendrites. Rather, our findings establish a transport pathway that selectively and specifically routes SV proteins to axons.

## A direct and selective transport pathway for dendritic cargo

To increase rigor, we conducted control experiments to determine whether our assay can, in fact, accurately identify dendritic transport. For this, we chose the transferrin receptor (TfR), a protein localized to dendrites (*Burack et al., 2000*; *Li et al., 2016a*; *West et al., 1997a*; *Farías et al., 2015*), and expressed it in cells with a bicistronic hook-reporter RUSH plasmid (*Figure 2—figure supplement*

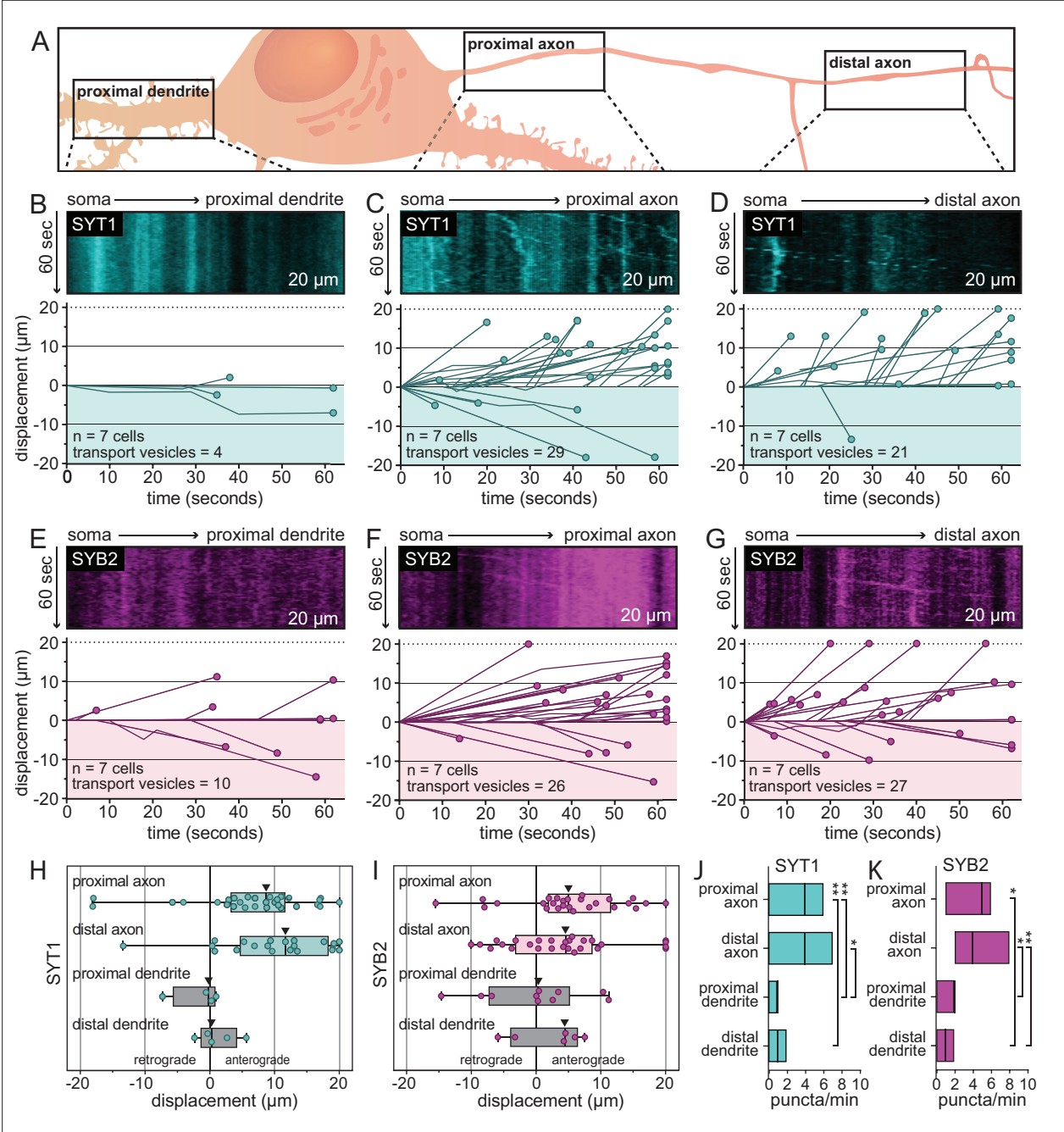

**Figure 2.** A direct and selective axonal transport pathway for SYT1 and SYB2 in rat hippocampal neurons. (**A**) Illustration outlining the proximal and distal regions that were imaged for each neuron. (**B**) Representative kymographs from the proximal dendrite of 14–16 days in vitro (DIV) rat hippocampal neurons after release of the tethered SYT1 reporter, revealing an absence of SYT1-bearing mobile organelles. For panels (**B–G**), all data were quantified and plotted immediately below the kymographs; the number of cells and transport vesicles are also indicated. (**C, D**) Representative kymographs from proximal and distal axons showing robust movement of the released SYT1 reporter, suggesting a direct axonal trafficking pathway. (**E–G**) Same as for panels (**B–D**) but using neurons expressing the SYB2 reporter. Displacement of transport vesicles containing the SYT1 (**H**) or SYB2 reporters (**I**) is plotted in the anterograde (positive) or retrograde (negative) direction with respect to the soma; arrowheads indicate median values. Both synaptic vesicle (SV) proteins are primarily trafficked in the anterograde direction. Mean values and descriptive statistics are found in *Figure 2—source data 1A*. (**J**) Flux of the SYT1-bearing transport vesicles, in the indicated compartments, are plotted as floating bars (min to max), line indicates median value. Data were collected from seven cells. A one-way ANOVA with multiple comparisons was run; p-values were as follows: proximal axon vs. distal axon = 0.56; proximal axon vs. proximal dendrite = 0.0021; proximal axon vs. distal dendrite = 0.0047; distal axon vs. proximal dendrite = 0.046; distal axon vs. distal dendrite = 0.091; proximal dendrite vs. distal dendrite = 0.99. (**K**) Same as panel (**J**), but for the SYB2 reporter. Data were collected from seven cells. Statistical tests were run as in panel (**J**) and p-values were as follows: proximal axon vs. distal axon = 0.998; proximal axon vs. proximal dendrite = 0.058;

*Figure 2 continued on next page*

*Figure 2 continued*

proximal axon vs. distal dendrite = 0.013; distal axon vs. proximal dendrite = 0.018; distal axon vs. distal dendrite = 0.0088; proximal dendrite vs. distal dendrite = 0.907. Mean values and descriptive statistics are found in *Figure 2—source data 1B*.

The online version of this article includes the following source data and figure supplement(s) for figure 2:

**Source data 1.** Descriptive statistics corresponding to *Figure 2*.

**Figure supplement 1.** Kymograph analysis.

**Figure supplement 2.** Representative kymographs from the distal dendrite of rat hippocampal neurons at 14–16 days in vitro (DIV) expressing the SYT1 (**A**) or SYB2 (**B**) reporter.

**Figure supplement 3.** SYB2 transport is slowed in proximal axons.

**Figure supplement 4.** Dendritic cargo is delivered to dendrites without passing through axons.

*4A*; *Chen et al., 2017*). Using the same criteria as *Figure 2* to select proximal and distal regions (*Figure 2—figure supplement 4B*), we observed that TfR was overwhelmingly trafficked to dendrites without passing through axons (*Figure 2—figure supplement 4C–F*), even when purposefully over-expressed. Indeed, there was some difficulty visualizing this construct, so it was expressed at higher levels. We note that there was a moderate population of non-moving TfR puncta observed in our experiments. We also note that the TfR reporter was subject to minor leakage, so some initial egress was missed. Hence, the non-moving puncta may represent the steady-state accumulation of this protein at its normal destination. Regardless, the majority of transport vesicles were found to be present in dendrites. These results demonstrate that this assay can, indeed, reveal dendritic targeting, as reported previously (*Farías et al., 2015*). This observation further validates our findings of direct and specific transport of SYT1 and SYB2 to axons.

## The direct and selective transport of SV proteins is obscured by overexpression

As stated in the Introduction, the expression levels of SV proteins can affect their localization; namely, overexpression results in the spillover of these proteins into other compartments, including the PM (*Marks et al., 1996*; *Pennuto et al., 2003*). Additionally, catch-and-release assays can cause mislocalization by overwhelming the early secretory pathway upon the bulk release of protein (*Adams et al., 2019*). Indeed, in agreement with previous studies, when we drastically overexpressed SYT1 via transfection, it spilled over into dendrites and "coated" the PM of all neurites; it also caused the growth of filopodia-like structures in the somatodendritic domain (*Figure 3A*; *Feany et al., 1993b*).

To formally address these concerns, we examined the effects of overexpression of the SYT1 and SYB2 reporters. We repeated the minimal expression paradigm used above (0.1x viral titer), and compared the subcellular distributions of these reporters when expressed at intermediate (1x viral titer) and high (10x viral titer) levels. Western blot analysis confirmed the overexpression of each protein, as compared to their endogenous counterparts (*Figure 3—figure supplement 1*); at lower virus titers, drastically fewer cells were transduced, so the exogenously expressed proteins escaped detection. We employed a quantitative ICC approach where the fluorescence intensity of antibodies against SYT1 or SYB2 were compared at synapses containing tagged and native protein, or just native protein. This analysis revealed that the low, intermediate, and high expression levels resulted in 1.14-, 1.22-, and 2.48-fold increases over the wild type (WT) SYT1 levels at the synapse, respectively, and 0.93-, 1.17-, and 1.87-fold changes for SYB2 (*Figure 3B and C*). In addition to changes in expression at individual synapses, increasing virus titer also resulted in increased transduction coverage (*Figure 3D and E*).

In addition to the relatively modest overexpression observed at individual synapses, both the SYT1 and SYB2 reporters also appeared throughout the PM of cells and spilled over into internal structures in dendrites at high expression levels (*Figure 3F and G*). A similar trend was observed at intermediate expression levels, albeit with lower signals in dendrites. Only when expression levels were low—effectively indistinguishable from the endogenous protein—and the dimmest cells were selected, did we observe the polarized, presynaptic distribution of the SYT1 and SYB2 reporters, at steady state, that is characteristic of SV proteins (*Figure 3F and G*; *Chapman, 2008*; *Südhof and Rothman, 2009*). We note that, under this low-expression paradigm, neurons are quite dim and, thus, challenging to image (i.e., required sensitive microscopy and a concentrated release of protein via the RUSH assay

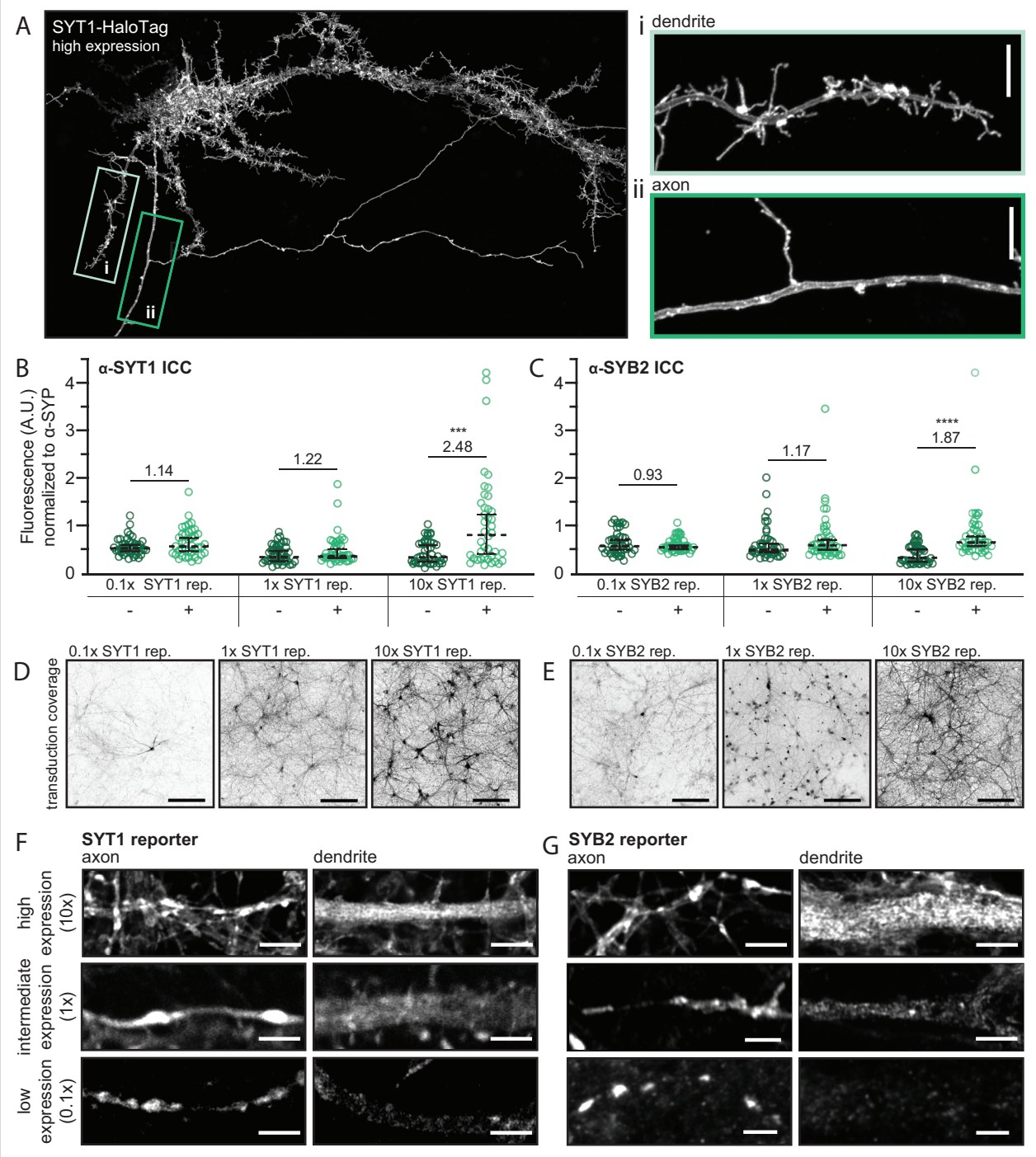

**Figure 3.** The direct and selective transport of synaptic vesicle (SV) proteins is obscured by overexpression. (**A**) A super-resolution MAX-projection image of 15 days in vitro (DIV) rat neurons expressing SYT1-HaloTag at high levels, as evidenced by the localization of SYT1-HaloTag in both axons and dendrites, along with the formation of filopodia-like structures in the somatodendritic compartment. The construct is visualized with JF549 HaloTag ligand. The boxed regions were expanded to show that the overexpressed protein accumulates on both the dendritic (**i**) and axonal (**ii**) plasma membrane (PM). Scale bar is 5 μm. (**B**) Graphs comparing the fluorescence intensity of the α-SYT1 antibody at synapses with or without tagged SYT1 reporter at low, intermediate, and high expression levels. We note that here, "x" represents the titer of virus (see *Figure 3—figure supplement 1* for comparison of wild type (WT) vs. transduced protein) where 1x is slightly less than endogenous levels; average relative expression levels are shown in panels (**B**) and (**C**). Data were plotted as median with 95% confidence intervals and Mann-Whitney tests were run comparing native protein to native and tagged protein for each virus dose (p-value$_{0.1x}$ = 0.36, p-value$_{1x}$ = 0.20, p-value$_{10x}$ = 0.0004). Average relative expression levels are included on the graph. (**C**) The same as panel (**B**) but comparing the fluorescence intensity of SYB2 with and without expression of the SYB2 reporter. Data were analyzed as in panel (**B**) (p-value$_{0.1x}$ = 0.79, p-value$_{1x}$ = 0.17, p-value$_{10x}$ = <0.0001). Data from panels (**B**) and (**C**) represent 40 synapses per condition, collected from four

*Figure 3 continued on next page*

*Figure 3 continued*

total fields of view from two different litters. Mean values and descriptive statistics are found in *Figure 3—source data 1*. (**D**) MAX-projection images of 14–16 DIV rat neurons expressing the SYT1 reporter showing the transduction coverage achieved with each viral dose. Scale bar represents 150 μm. Images were adjusted individually, with linear brightness and contrast, to the brightest area of the image to aid in visualization. (**E**) The same as panel (**D**), but for the SYB2 reporter. Super-resolution optical sections of the SYT1 (**F**) and SYB2 (**G**) reporters at low, intermediate, and high expression levels, in axons and dendrites, demonstrate that as expression levels increase, SV proteins spillover into dendrites. Scale bar represents 2.5 μm. Corresponding axon and dendrite images, at each expression level, were adjusted with the same linear brightness and contrast settings.

The online version of this article includes the following source data and figure supplement(s) for figure 3:

**Source data 1.** Descriptive statistics corresponding to *Figure 3*.

**Figure supplement 1.** Expression levels of the SYT1 and SYB2 reporters as compared to native protein.

to observe transport), which may explain why higher expression levels are often employed in axonal transport studies.

The spillover of exogenously expressed SV proteins into dendrites at high, and even intermediate expression levels, at steady state (*Figure 3A, F and G*) is likely due to indiscriminate transport when expression levels are not carefully controlled. Therefore, it is imperative to use a low-expression paradigm to study the native trafficking pathway utilized by these proteins. Indeed, a mere doubling of endogenous protein at synapses corresponded with the widespread mistargeting of SV proteins to dendrites and the PM. We note that a similar trend of protein mislocalization at high expression levels was observed using a transfection approach (*Figure 3A*); only when we 'diluted' the plasmid of interest, by mixing and co-transfecting it with a dummy plasmid, were we able to minimize overexpression artifacts (note: this approach was used in Figure 5, discussed below). Simply reducing the total amount of the plasmid of interest for transfection was not sufficient to mitigate the rampant mistargeting. Taken together, these data demonstrate the extent to which overexpression can cause SV proteins to be mistargeted, at moderately low levels of overexpression, to ultimately obscure their native transport pathway. Additionally, these results help to reconcile the discrepancies between the current study and previous studies reporting the trafficking of proteins, like SYB2, in both axons and dendrites (*Nabb and Bentley, 2022*; *Sampo et al., 2003*).

## Molecular determinants that underlie the polarized transport of SYT1 to axons

Next, we sought to uncover how SYT1 is selectively sorted to axons. Although sorting motifs are not yet defined for this protein, it is both palmitoylated (*Chapman et al., 1996*; *Heindel et al., 2003*) and glycosylated (*Perin et al., 1991*), and both modifications have been proposed to play roles in its trafficking (*Kang et al., 2004*; *Han et al., 2004*; *Atiya-Nasagi et al., 2005*; but see also *Kwon and Chapman, 2012*). We addressed this idea in our transport assay by mutating all five putative palmitoylation sites, and all three glycosylation sites, of SYT1 to prevent these modifications (*Figure 4A*). Hereafter, this mutant is referred to as the SYT1 palmitoylation/glycosylation mutant, or SYT1-PGM. In parallel, we assessed the role of the tandem C2-domains of SYT1, which sense $Ca^{2+}$ and interact with a variety of effectors, in targeting this SV protein to nerve terminals. We note that deletion of both C2-domains caused the truncated protein to be marooned on the PM (*Courtney et al., 2019*). However, how this deletion mutant, termed SYT1ΔC2AB, is sorted and transported remained unknown, so we characterized it using the RUSH assay (*Figure 4A*). These experiments were conducted in a SYT1 knockout background to avoid potential homomeric interactions with endogenous SYT1 (*Brose et al., 1992*; *Courtney et al., 2021*; *Perin et al., 1991*), wherein mutant protein could "piggyback" onto the native protein in the secretory pathway and obscure potential transport defects associated with the mutant protein (*Figure 4B*).

Consistent with the experiments above (*Figure 1*, *Figure 2*, and *Figure 1—figure supplement 3*), the endpoint targeting of each reporter construct was established at low expression levels. The full-length SYT1 reporter was included as a positive control and was correctly targeted to synapses, as confirmed by its colocalization with the synaptic marker, synaptophysin (SYP) (*Figure 4C*). The SYT1-PGM construct accumulated in the soma as well as the axonal compartment, where it was colocalized with SYP; it was virtually undetectable in dendrites, similar to our findings using the WT SYT1 reporter (*Figure 4D*). The truncated SYT1 protein, SYT1ΔC2AB, was present throughout axons, likely

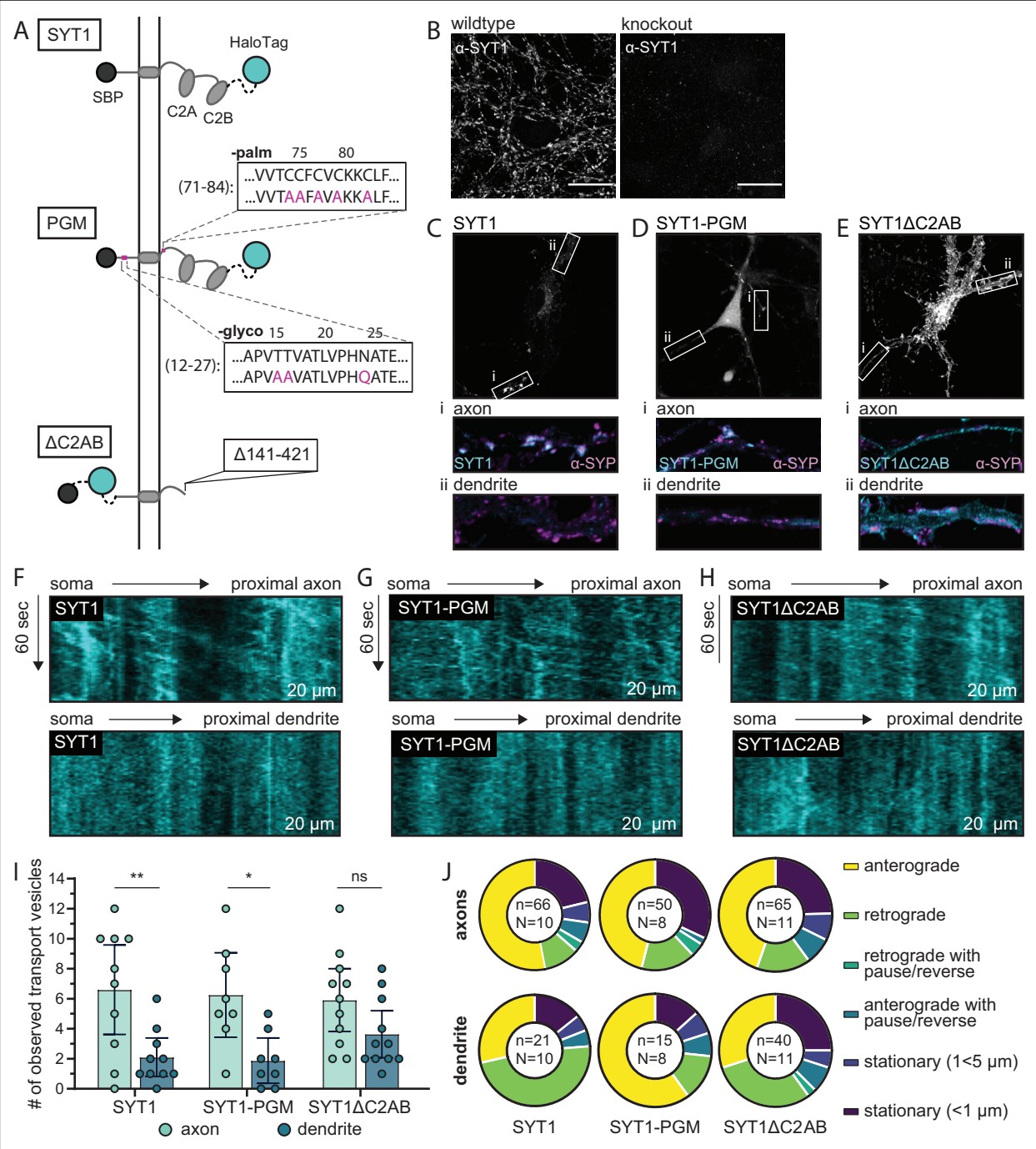

**Figure 4.** Molecular determinants that underlie the polarized transport of SYT1 to axons in mouse hippocampal neurons. (**A**) Illustration of retention using selective hooks (RUSH) reporters used for these experiments: wild type (WT) SYT1 reporter, the SYT1 palmitoylation and glycosylation mutant (SYT1-PGM), and SYT1 truncated after position 140 (SYT1ΔC2AB). Each construct has a HaloTag for visualization. (**B**) ICC confirms the knockout of endogenous SYT1. For WT and knockout conditions, identical laser and gain settings were used. Scale bar represents 20 µm. (**C–E**) The endpoint localization of WT SYT1, SYT1-PGM, and SYT1ΔC2AB was visualized by labeling the appended HaloTag with JF549. The boxed regions were expanded and are shown below each panel to better reveal the localization of each construct in axons (**i**) and dendrites (**ii**), as compared to the α-SYP ICC signals. Note that all neurons were immunostained for SYP, but only a handful of cells expressed each SYT1 construct. ICC images were adjusted to the brightest area of the image to aid in visualization. All settings were kept consistent between corresponding axon/dendrite insets for a given cell and condition, and all images (**B–E**) were adjusted with linear brightness and contrast. Representative kymographs from proximal axons showing robust anterograde movement of the released SYT1 (**F**), SYT1-PGM (**G**), and SYT1ΔC2AB (**H**) reporters as compared to dendrites, demonstrating selective trafficking of WT and SYT1-PGM, but not SYT1ΔC2AB, to axons. (**I**) The number of transport vesicles was plotted for each construct as the mean with 95% CI. A one-way

*Figure 4 continued on next page*

*Figure 4 continued*

ANOVA was run (p=0.0008), and a Šídák's multiple comparisons test was used to compare transport in axons and dendrites of all three RUSH reporters. Significant differences in axonal vs. dendritic transport were observed for WT SYT1 (p=0.0097) and SYT-PGM (p=0.036), indicating polarized trafficking. In contrast, the transport of SYT1ΔC2AB was not significantly polarized (p=0.49). A complete list of multiple comparisons results can be found in *Figure 4—source data 1*. Data were collected for 10 cells (SYT1), 8 cells (SYT1-PGM), or 11 cells (SYT1ΔC2AB), from four litters. Mean values and descriptive statistics are found in *Figure 4—source data 2*. (**J**) The movement of each transport vesicle categorized as anterograde, retrograde, retrograde with pause/reverse, anterograde with pause/reverse, stationary (1<5 µm), or stationary (<1 µm) and plotted as a fraction of the total number of transport vesicles observed for each compartment, for each construct. The total number of (n) transport vesicles from (N) cells are indicated. Exact fractions can be found in *Figure 4—source data 3*.

The online version of this article includes the following source data and figure supplement(s) for figure 4:

**Source data 1.** Šídák's multiple comparisons test results corresponding to *Figure 4I*.

**Source data 2.** Descriptive statistics corresponding to *Figure 4I*.

**Source data 3.** Transport vesicle movement analysis (fraction of a whole) related to *Figure 4J*.

**Figure supplement 1.** The SYT1ΔC2AB reporter is present on the plasma membrane.

on the PM (*Figure 4—figure supplement 1*; *Courtney et al., 2019*), and was also observed in the somatodendritic compartment, indicating mistargeting of the protein (*Figure 4E*).

Next, we studied the transport of the PGM and ΔC2AB mutants using the RUSH system. Each fusion protein was successfully sequestered in the ER and was released upon the addition of biotin. Notably, both mutants successfully left the Golgi in transport vesicles and did not immediately fuse with the somatic PM, but instead were trafficked into neurites. We also note that these experiments were conducted without the addition of ER-targeted GFP, because the RUSH assay workflow improved when cells only expressed the hook and the reporter (i.e., constructs released more reliably, and the post-Golgi vesicles were brighter and easier to visualize).

In knockout neurons, the WT SYT1 reporter was—again—trafficked in a polarized manner to axons (*Figure 4F and I*). In contrast to previous studies (*Han et al., 2004*; *Kang et al., 2004*; *Atiya-Nasagi et al., 2005*), but consistent with our findings under steady state, SYT1-PGM was selectively trafficked to axons, consistent with our observations of the WT protein (*Figure 4G and I*). Conversely, the SYT1ΔC2AB construct entered axons and dendrites at similar rates, with a non-significant trend toward preferential entrance into axons (p=0.49) (*Figure 4H and I*). Interestingly, the C2AB deletion mutant resulted in increased dendritic transport as compared to the WT protein, while axonal transport remained unchanged, indicating these domains might play a role in targeting SYT1 to different subsets of transport vesicles with distinct destinations. Next, the movements of each transport vesicle in axons and dendrites, for each of the three constructs, were quantified (*Figure 4J* and *Figure 4—source data 3*). For all three constructs, the majority of transport vesicles proceeded in an anterograde direction in axons. In dendrites, the majority of mobile puncta carrying SYT1 and SYT1ΔC2AB moved in a retrograde direction, though a considerable fraction of SYT1ΔC2AB vesicles were stationary. These mobile vesicles either represent protein that is moving from the dendritic ER to toward the soma, or are transport vesicles that egressed prior to imaging and are moving in a retrograde direction at the time the imaging was conducted. Interestingly, SYT1-PGM overwhelmingly moved in an anterograde direction in dendrites under the non-equilibrium conditions of these experiments. However, the total number of transport vesicles carrying SYT1 and SYT1-PGM in dendrites was relatively low, so the observed differences should be interpreted with caution. Taken together, these experiments reveal that the tandem C2-domains play a role in the proper targeting of SYT1. In contrast, palmitoylation and glycosylation were dispensable for selective targeting of SYT1 to axons.

## Transport vesicles deliver SV proteins to the presynaptic PM, creating a depot for SV biogenesis

Finally, we sought to visualize the immediate destination of newly delivered SV proteins after they are sorted to axons. Previous studies have established that SVs are assembled at the presynapse (*Buckley et al., 2000*; *Nakata et al., 1998*; *Okada et al., 1995*; *West et al., 1997b*), but how they are first generated in that compartment remains unknown. It has long been hypothesized that SV proteins, prior to their incorporation onto nascent SVs, are first delivered to the presynaptic PM as the final destination of their maiden voyage to synapses (*Buckley et al., 2000*; *Feany and Buckley, 1993a*;

*Hannah et al., 1999*; *Régnier-Vigouroux et al., 1991*). However, this idea stems from experiments done in PC12 cells and CHO fibroblasts, which do not contain SVs. Since some SYT1 and SYB2 molecules are present on the PM at steady state (*Sankaranarayanan and Ryan, 2000*), and overexpressed protein also accumulates on the PM (*Figure 3*), it is reasonable to postulate that SV precursors are initially trafficked through the PM of mammalian neurons as a necessary part of their life cycle.

We addressed this longstanding question by developing a novel HaloTag labeling approach to conduct pulse-chase experiments using permeant and non-permeant Janelia Fluor (JF) HaloTag ligands (HTL) (*Grimm et al., 2015*). To assess whether these SV proteins are delivered directly to the presynaptic PM, we appended a HaloTag to the intravesicular terminus of SYT1 and SYB2 (termed HaloTag-SYT1 and SYB2-HaloTag) so that the tag is exposed to the outside of the cell when the SV protein is incorporated into the PM (*Figure 5A*). These constructs were sparsely co-transfected with an SYP-GFP fusion protein to mark synapses. The ratios of SYT1, or SYB2, to the SYP plasmid in these co-transfection experiments were determined experimentally so that the HaloTagged protein could be visualized, but was minimally expressed via concurrent dilution with the SYP plasmid. Immediately after co-transfection, neurons were grown with or without JF549i, a non-permeant fluorescent HTL, in the culture media. Impermeability of the ligand, under our experimental conditions, was confirmed empirically (*Figure 5—figure supplement 1*). This HTL labeled any copies of tagged SYT1 or SYB2 that passed through the PM (*Xie et al., 2017*). If SYT1 and SYB2 were delivered to a presynaptic sorting compartment, rather than the PM, they would not be labeled with the non-permeant HTL (*Figure 5B*). After 6 days the degree of labeling with JF549i was assessed via imaging. Then, the neurons were challenged using a permeant ligand, JF549, which has nearly the same structure, and fluorescence properties, as JF549i. Subsequent incubation with the permeant HTL labeled, and hence revealed, any remaining unlabeled protein that did not pass through the PM. This labeling scheme is illustrated in *Figure 5B and C*.

Synaptic activity in our cultures, and hence the recycling of SVs, could contribute to SV proteins passing through the PM. To minimize this potential confound, the light chain of tetanus toxin (TeTx-LC) was co-expressed, using lentivirus, to cleave endogenous SYB2 and inhibit synaptic activity and SV recycling (*Figure 5D*; *Figure 5—figure supplement 2*; *Bao et al., 2018*; *Schiavo et al., 1992*). The tagged SYB2 construct was mutated at residues 76 and 77 (Q76V and F77W) to make it resistant to this toxin (*Schiavo et al., 1992*). Cleavage of endogenous SYB2 by TeTx-LC did not affect the expression of other canonical SV proteins (*Figure 5D*).

To quantify labeling of HaloTag-SYT1 and SYB2-HaloTag at the presynaptic PM, the co-transfected SYP-GFP was used to define individual synapses, and fluorescence intensity was measured at each synapse before and after addition of the permeant HTL, JF549. As expected, in cultures grown in the absence of JF549i, we observed a dramatic increase in the fluorescence intensity at synapses upon the addition of permeant JF549 for both SYT1 and SYB2. However, synapses cultured with JF549i for 6 days exhibited minimal changes in fluorescence after addition of permeant JF549. These findings reveal that the majority of tagged SYT1 and SYB2 were already labeled with the membrane impermeant HTL. Hence, the majority of newly delivered SYT1 and SYB2 molecules pass through the PM, independent of synaptic activity (*Figure 5E and F*). We cannot rule out that some tagged protein was delivered to an internal compartment, however, the all-or-nothing labeling we observed with the non-permeant ligand gives no indication of an internal depot that was protected from the non-permeant dye. Additionally, it is unlikely that the residual minis that occur in the presence of TeTx-LC (5%) contribute significantly to labeling at the PM for two reasons. Namely, in the absence of activity, the SV cycle and SV reformation are stalled, so tagged protein is unlikely to be efficiently incorporated into newly produced, fusogenic vesicles that are able to participate in spontaneous or evoked release. Second, if tagged protein was delivered to an internal compartment, only to be subsequently labeled at the PM, this would require a fast and efficient pathway for incorporation into fusion-competent vesicles that undergo spontaneous release. However, we have conducted preliminary experiments using RUSH to rescue synaptic neurotransmission in SYT1 KO neurons and found that incorporation of tagged protein into functional vesicles takes days. This is consistent with the model, alluded to above, in which SV recycling drives incorporation of newly delivered proteins into SVs. While we cannot rule out that a small fraction of tagged protein could be labeled through the residual minis that occur in the presence of TeTx-LC, this is unlikely to contribute to a significant degree. Thus, we conclude the major pathway involves delivery of newly synthesized SV proteins to the PM. We emphasize that these

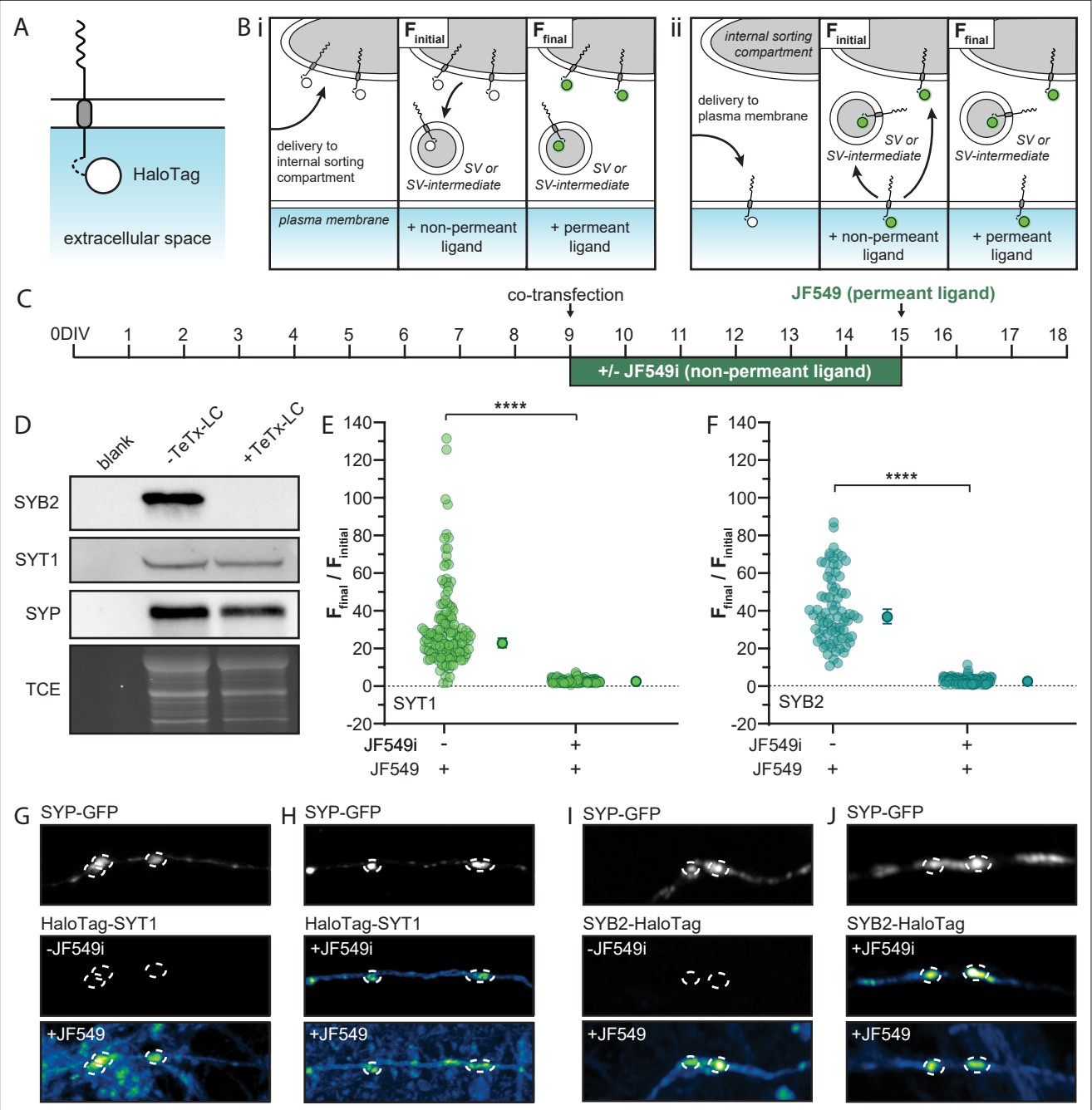

**Figure 5.** Transport vesicles deliver synaptic vesicle (SV) proteins to the presynaptic plasma membrane (PM) in rat hippocampal neurons, creating a depot for SV biogenesis. (**A**) Illustration of a generic integral membrane protein (representing SYT1 and SYB2) with a luminal HaloTag to allow for selective labeling at the PM. (**B**) Schematic of the HaloTag ligand (HTL) labeling protocol, shown within a nerve terminal, with the non-permeant ligand incubation step to label surface protein. If SYT1 and SYB2 are first delivered to an internal sorting compartment (**i**), rather than the PM (**ii**) prior to SV or SV-intermediate formation, they will not pass through the PM and so will not be labeled by the non-permeant ligand (green). Incubation with permeant ligand labels the remaining tagged protein and the resulting change in fluorescence denotes the efficiency of PM delivery. The signal from labeling with the non-permeant ligand was referred to as $F_{initial}$, where the unlabeled control coverslips still yielded a small background signal, producing a reproducible non-zero value that allowed us to calculate ratios. The subsequent signal after labeling with permeant ligand was called $F_{final}$. This labeling step included unbound ligand which, while weak and diffuse, results in a slight increase in the background signal. To counteract this, ROIs were drawn to include only the fluorescence intensity within the synapse. (**C**) Timeline for the transfection and labeling protocols. Briefly, cultured rat hippocampal neurons were transduced with TeTx-LC virus on 5 days in vitro (DIV) and then co-transfected on 9 DIV with HaloTag-SYT1 or SYB2-HaloTag, and SYP-GFP; the GFP construct was included to mark synapses and 'dilute' the HaloTag plasmid to achieve lower expression levels. Half of the coverslips were incubated in non-permeant HTL (JF549i) immediately after co-transfection to label any tagged protein that was delivered to

*Figure 5 continued on next page*

*Figure 5 continued*

the PM. Six days later (15 DIV) neurons were rinsed, imaged, and incubated with permeant ligand (JF549), to label any remaining tagged protein, and imaged again. (**D**) Immunoblot of cells transduced with a virus expressing TeTx-LC, resulting in the cleavage of endogenous SYB2 and the inhibition of SV recycling. We note that the SYB2 fusion protein used in these experiments harbored two point mutations to render it resistant to TeTx-LC (see Methods). Blots were probed for endogenous SYB2, SYT1, and SYP, with a TCE loading control. The normalized ($F_{final}/F_{initial}$) change in fluorescence intensity of the SYT1 (**E**) and SYB2 (**F**) fusion proteins upon adding permeant fluorescent ligand to cells grown with or without non-permeant ligand for 6 days; mean values with 95% CI are plotted to the right of each scatter plot. Data were analyzed with unpaired t-tests for both proteins; p-values = <0.0001. Panel (**E**) contains data from 156 synapses cultured in the presence of JF549i, and 136 synapses grown in the absence of this HTL. Data for both groups were from 8 fields of view from 4 different litters. Panel (**F**) contains data from 107 synapses cultured in the presence of JF549i, and 79 synapses grown in the absence of this HTL. Data for both groups were from five fields of view from three different litters. Mean values and descriptive statistics for SYT1 and SYB2 can be found in *Figure 5—source data 1*. Panels (**G, H**) are representative images of SYP-GFP to mark synapses (dashed circles), and the corresponding HaloTag-SYT1 signals under the indicated conditions; in the bottom panels the JF549 ligand was not washed away, resulting in a higher background. For all conditions, identical laser and gain settings were used. Any linear brightness and contrast adjustments were applied to all conditions. (**I, J**) Same as panels (**G**) and (**H**), but for SYB2-HaloTag.

The online version of this article includes the following source data and figure supplement(s) for figure 5:

**Source data 1.** Descriptive statistics corresponding to *Figure 5*.

**Figure supplement 1.** JF549i HaloTag ligand is not cell-permeant after six days.

**Figure supplement 2.** Expression of TeTx-LC disrupts synaptic activity.

results were readily apparent by eye, as shown in the representative images of cultures grown with and without the non-permeant ligand and chased with a permeant ligand (*Figure 5G–J*). In light of these findings, we propose that delivery of SYT1 and SYB2 to the presynaptic PM creates a depot from which SV biogenesis can occur.

## Discussion

A neuron's ability to sort proteins and transport cargo to synapses underlies the function of the nervous system and is a process that is maintained throughout the lifespan of the cell. As such, several theories have been proposed for how transport occurs. For the past two decades, the leading model posits that axonal proteins are delivered indiscriminately to all neurites and are subsequently selectively retained in axons (*Bentley and Banker, 2016*; *Sampo et al., 2003*). In sharp contrast, the transport of dendritic cargos has been shown to be selective, and vesicles carrying dendritic cargo are trafficked directly to dendrites without entering axons. This presents an apparent paradox, because both dendritic and axonal arbors can have elaborate morphologies. Transporting the cargos, destined for axons, through another exceptionally complex compartment would further complicate this sorting process. Consequently, selective retention, and other variants of this non-selective transport theory, appeared to be a high-effort, low-reward method of establishing and maintaining polarity. While inefficient mechanisms cannot be ruled out, the development of a specific postal system for dendrites, but not axons, remained somewhat puzzling. Using modern approaches and—importantly—by carefully controlling protein expression levels, our data sharply contrast the selective retention model and reveal that members from two distinct families of SV proteins are directly and specifically routed to axons.

The establishment of neuronal polarity and its maintenance has been studied for years, which prompts the question of why the direct and selective transport of SV proteins is only now being characterized. As we show, without the careful control of expression levels, direct transport is obscured by the spillover and mistargeting of cargo into other compartments of neurons. We note that SV proteins seem to be particularly susceptible to this artifact, as the intentional overexpression of TfR did not appear to affect its selective transport to dendrites. In earlier studies, axonal proteins were likely delivered indiscriminately to axons and dendrites due to spillover and mistargeting, as a consequence of overexpression. Furthermore, there has been confusion in the field about what constitutes direct transport. Some groups defined direct transport according to where post-Golgi vesicles initially fuse (*Nabb and Bentley, 2022*; *Sampo et al., 2003*). By this definition, axonal cargo can leave the Golgi, traverse the entire neuron, including through dendrites, fuse in axons, and still be considered a direct transport pathway. So, although it has been considered a distinct model, this "direct" transport pathway remains a version of selective retention.

Prior to the current study, there were reports in the literature that suggested a direct and selective transport pathway for SV proteins. Namely, a rigorous study focusing on nematode DA9 bipolar neurons revealed that SYB2 was delivered directly to presynaptic boutons (*Li et al., 2016b*). However, these neurons have a more simplified microtubule organization than mammalian hippocampal neurons, so further studies were necessary to confirm whether this pathway extended to the complex cytoskeleton of mammalian cells. Experiments using dorsal root ganglion cells from mouse and rat suggested MAP2-dependent selective cargo sorting and transport of axonal proteins (*Gumy et al., 2017*), but these sensory neurons have a pseudounipolar morphology, meaning they have a single bifurcated axon and no dendrite, and they lack an axon initial segment, making them distinct from hippocampal neurons. A more recent study, using mouse neurons, indicated that axonal proteins do not enter dendrites (*Karasmanis et al., 2018*). However, while dendritic exclusion was clearly established, entry into axons was not shown. Nevertheless, these papers began to question the idea of non-polarized transport.

Newer approaches have also made it possible to directly address the trafficking of axonal cargos as they egress from the soma (*Boncompain et al., 2012*; *Grimm et al., 2017*; *Los et al., 2008*). At the outset of the current study, we showed that when expression levels are carefully controlled, two topologically distinct SV proteins egressed from the soma directly to axons, thus uncovering a novel selective membrane transport pathway. A small fraction of mobile puncta were detected in dendrites, so it is possible that the fidelity of axonal targeting is not absolute. However, there are other possible explanations for this observation, namely, even very mild overexpression causes a small degree of spillover into dendrites, trafficking is altered by the addition of the fusion moiety, or these puncta represent protein in the dendritic ER that is moving toward the Golgi in the soma. Regardless, transport was highly selective for axons vs. dendrites. Another issue is that dendrites are larger than axons, so the number of transport vesicles are potentially under sampled and underestimated in this compartment. However, videos were analyzed such that activity across the full width of each neurite was included in the kymograph, to take—in part—differences in neurite size into account. We also argue that the few vesicles we observed in dendrites tend to move in the z-dimension, so we believe transport vesicles are not being missed, rather it is more likely that we are underestimating their displacement as they traffic in and out of the imaging plane. We also note that the selective targeting of SYT1 appears to be more precise than that of SYB2. Whether this is due to SYB2 being intrinsically more promiscuous or is an artifact resulting from its fusion to HaloTag or SBP remains unclear. Regardless, the findings reported here reveal strongly biased direct and selective transport of both SYT1 and SYB2 to axons.

We also used the RUSH assay to conduct structure-function studies of SYT1 and found that glycosylation and palmitoylation were dispensable for direct transport to axons. In contrast, removing the C2-domains of SYT1 did not affect axonal transport, but rather increased transport into dendrites, thereby disrupting the polarized distribution of this protein. As mentioned under Results, these findings suggest that the tandem C2-domains might act to suppress dendritic transport. This raises the possibility that these domains help to direct SYT1 to transport organelles that are specific to axons, while the truncated protein is targeted to vesicles that do not undergo polarized transport. Additionally, this deletion mutant was present throughout the plasmalemma of both axons and dendrites at steady state. Clearly, further study is needed; the deletion mutant impairs the polarized transport and distribution of SYT1, but there is still a trend toward axonal enrichment. Nevertheless, these initial findings point to a role for the C2-domains in polarized transport of this protein. We note that the HaloTag reporter was appended to the N-terminus of the truncation mutant but was on the C-terminus of the full-length protein and the PGM mutant. However, it is established that, after careful design, tags at either end of the full-length protein are tolerated (*Diril et al., 2006*; *Vevea and Chapman, 2020*), so this is unlikely to affect localization (*Figure 4*, *Figure 5*).

This study also addressed the first half of the life cycle of SV proteins by conducting pulse-chase HaloTag assays to answer the long-standing question of whether SYT1 and SYB2 are—in fact—first delivered to the synaptic PM or presynaptic endosomes. Our results strongly argue that both of these proteins are delivered to the presynaptic PM where they serve as a reservoir from which SVs are eventually created. This initial fusion reaction is potentially mediated via tetanus-insensitive VAMP, called VAMP7 (*Galli et al., 1998*; *Chaineau et al., 2009*). Then, during normal recycling, SYT1 is internalized via its tandem C2-domains, potentially via mechanisms that mediate SV retrieval from the PM (*Courtney et al., 2019*; *Jarousse et al., 2003*), and efficient retrieval of SYB2 is mediated by its

interaction with SYP (*Gordon et al., 2011*; *Harper et al., 2021*). Retrieval may involve interactions with various adaptor proteins, but the emerging view is that these interactions are unlikely to occur at the PM, as clathrin-mediated endocytosis is no longer thought to mediate the internalization of SV proteins (*Watanabe et al., 2013*). Regardless, these pulse-chase experiments reveal the first step in the biogenesis of SVs: selective delivery and incorporation of SYT1 and SYB2 in the presynaptic PM, as proposed decades ago (*Buckley et al., 2000*; *Feany and Buckley, 1993a*; *Hannah et al., 1999*; *Régnier-Vigouroux et al., 1991*).

A key issue moving forward is to understand the cargo selection process that underlies axon-specific transport, and to further understand how newly delivered proteins are incorporated into SVs. Finally, a complete picture will not emerge until the other half of the life cycle of SV proteins is understood, namely how aged proteins are selected for, and undergo, degradation (*Birdsall and Waites, 2018*; *Cohen et al., 2013*; *Hoffmann-Conaway et al., 2020*; *Na et al., 2012*; *Sheehan et al., 2016*; *Sheehan and Waites, 2017*). New tools have made it possible to address these questions, including HaloTag pulse-chase approaches, in conjunction with organelle isolation and mass spectrometry. These techniques promise to reveal, in biochemical detail, the itinerary of SV proteins as they are created and destroyed.

## Methods
### Cell culture
Hippocampal neurons were dissected from pre-natal Sprague-Dawley rats on E18 (Envigo), or post-natal SYT1 conditional knockout floxed mice (*Quadros et al., 2017*) on P0-P1. Hippocampal tissue was maintained in chilled hibernate A media (BrainBits, HA) during dissection. After dissection, hippocampi were incubated in 0.25% trypsin (Corning, 25-053 CI) for 30 min at 37°C, triturated in Dulbecco's Modified Eagle Medium (DMEM) (Thermo Fisher Scientific, 11965-118) supplemented with 10% fetal bovine serum (Atlanta Biological, S11550H) plus penicillin-streptomycin (Thermo Fisher Scientific, MT-30-001 CI), to dissociate tissue. Rat neurons were plated on 18 mm coverslips Warner instruments, 64-0734 (CS-18R17) that had been coated with poly-D-lysine (Thermo Fisher Scientific, ICN10269491) for 1 hr at room temperature, at a density of 125,000 cells per coverslip, in supplemented DMEM. Mouse hippocampal neurons were also plated on 18 mm coverslips, but these were coated in poly-D-lysine and mouse laminin (Thermo Fisher Scientific, 23017015) for 2 hr at 37°C. For both rat and mouse neurons, once the cells had settled (<1 hr) DMEM was exchanged for Neurobasal-A Media (NBM) (Thermo Fisher Scientific, 10888-022) supplemented with N21-MAX Media Supplement (R&D Systems, AR008) (*Chen et al., 2008*), Glutamax (2 mM Gibco, 35050061), and penicillin-streptomycin. Additional supplemented NBM was added every 3–4 days to maintain the health of the cultures.

### Constructs
For the WT SYT1 (UniProt accession no. P21707) RUSH reporter, a pre-prolactin leader sequence and SBP were appended to the N-terminus, and a HaloTag (Promega, G7711) was fused to the C-terminus, of the SYT1 cDNA (*Figure 1B*). Each of these moieties, in this and all other constructs, were attached via a flexible GS(GSS)$_4$ linker. For the palmitoylation and glycosylation mutant form of the SYT1 reporter, the palmitoylation sites of SYT1, C74, C75, C77, C79, and C82 were substituted with Ala residues, and the glycosylation sites of SYT1, T15/T16, and N24 were substituted with Ala and Gln residues, respectively, using site-directed mutagenesis (Agilent Technologies, 210518). The truncated form of the SYT1 reporter, SYT1ΔC2AB (a.a. 1–140), was generated in the same manner as the full-length protein, except that the HaloTag was placed at the N-terminus of the SYT1 coding sequence. For the SYB2 (UniProt accession no. P63045) RUSH reporter, the HaloTag and SBP were appended to the C-terminus. For all SYT1 and SYB2 RUSH reporters, the HaloTag and the SBP were added in distinct positions to avoid steric interference between the SBP and the streptavidin hook. The streptavidin hook with an ER retention signal (Lys·Asp·Glu·Leu; KDEL) was made as a separate construct by sub-cloning it from Str-KDEL_neomycin, a gift from F Perez (Paris, France) (Addgene plasmid #65306; RRID:Addgene_65306) (*Boncompain et al., 2012*), into a pFUGW transfer plasmid (gift from D Baltimore [Pasadena, CA]; Addgene plasmid #14883; http://n2t.net/addgene:14883; RRID:Addgene_14883) (*Lois et al., 2002*). A FLAG tag (DYKDDDDK) was added to the C-terminus of all RUSH reporter constructs, immediately prior to the stop codon, to compare expression levels

between co-expressed proteins. The TfR construct was kindly provided by J Bonifacino (Bethesda, MD) (*Chen et al., 2017*).

For the pulse-chase studies, a non-RUSH HaloTag-SYT1 construct was generated using the same pre-prolactin leader sequence as above, but now followed by a HaloTag at the N-terminus of SYT1; for control experiments, the HaloTag was instead placed at the C-terminus. A non-RUSH SYB2-HaloTag construct was generated by appending a HaloTag to the C-terminus of the protein; the SYB2 cDNA harbored Q76V and F77W mutations to make it resistant to TeTx-LC. The TeTx-LC construct was subcloned from pGEMTEZ-TeTxLC, a gift from R Axel, J Gogos, and CR Yu (Addgene plasmid # 32640; http://n2t.net/addgene:32640; RRID:Addgene_32640) (*Yu et al., 2004*) into a pFUGW transfer plasmid (*Lois et al., 2002*). To mark synapses, a SYP GFP fusion protein (SYP-GFP), with the same flexible GS(GSS)$_4$ linker between the C-terminus of SYP and the GFP moiety, was used. All constructs were generated by overlap extension PCR and subcloned into the backbone using in-fusion cloning (Takara Bio, 638911). Constructs were sequenced fully, and all maxi-preps were re-sequenced prior to use.

## Constructs used in this study

pFsynW SYT1 reporter
pFsynW SYB2 reporter
pFsynW KDEL Hook
pFsynW SYT1-PGM reporter
pFsynW SYT1ΔC2AB reporter
pEF HaloTag-SYT1
pEF SYT1-HaloTag
pEF SYB2-HaloTag
pFsynW SYP-GFP
pFsynW TeTx-LC

## Lentivirus production and use

Relevant constructs were subcloned into a pFUGW transfer plasmid. To make lentiviral expression neuron-specific, the ubiquitin promoter was replaced with a human synapsin I promoter (*Kügler et al., 2003*). Lentiviral particles were generated via calcium phosphate co-transfection of HEK293T cells (ATCC, CRL-3216; RRID:CVCL_0063) at 30–40% confluency with the pFUGW transfer plasmid and the packaging plasmids, pCD/NL-BH*DDD and pLTR-G. Plasmids pCD/NL-BH*DDD (Addgene plasmid #17531; http://n2t.net/addgene:17531; RRID:Addgene_17531) (*Zhang et al., 2004*) and pLTR-G (Addgene plasmid #17532; http://n2t.net/addgene:17532; RRID:Addgene_17532) (*Reiser et al., 1996*) were gifts from J Reiser (Bethesda, MD). HEK293T cells were tested for mycoplasma contamination using the Universal Mycoplasma Detection Kit (ATCC; 30-1012K), validated using Short Tandem Repeat profiling by ATCC (ATCC; 135-XV), and maintained in DMEM supplemented with 10% FBS and penicillin-streptomycin. The supernatant was collected 48 hr after transfection, filtered with a 0.45 mm PVDF filter to remove cells and debris, and concentrated by ultracentrifugation at 110,000 × *g* for 2 hr. Viral particles were re-suspended in Ca$^{2+}$/Mg$^{2+}$-free phosphate-buffered saline (PBS), aliquoted, and stored at –80°C (*Kutner et al., 2009*).

For pulse-chase experiments, neurons were transduced with virus expressing TeTx-LC on 5 days in vitro (DIV). For RUSH release experiments, neurons were transduced with the streptavidin hook virus on 8 DIV and transduced with a reporter virus on 9 DIV. In *Figure 1* and *Figure 2*, a virus that expressed GFP with a 'KDEL' retention signal on the C terminus to label ER was also transduced on 9 DIV. Cells were imaged on 14–16 DIV. Lentivirus was titrated based on fluorescence and coverage unless otherwise stated in the text.

## Transfection

For HaloTag pulse-chase experiments, neurons were cultured in 12-well cell culture plates (Genesee Scientific; 25-106) and co-transfected with SYP-GFP and SYT1-HaloTag, HaloTag-SYT1, or SYB2-HaloTag, on 9 DIV using Lipofectamine LTX Reagent with PLUS Reagent (Thermo Fisher Scientific, 15338-100). Briefly, DNA plasmids were diluted in 25 μl Opti-MEM I Reduced Serum Medium (Gibco; 31985062), then 0.25 μl PLUS reagent was added. Separately, 1 μl LTX Reagent was diluted in 25 μl of

Opti-MEM I. The DNA-PLUS reagent mixture was added dropwise to the LTX reagent mixture, then added to culture media in each well.

## JF dye usage

HTL-conjugated JF dyes were graciously provided by L Lavis (Ashburn, VA). We made use of JF549 and JF549i. For protein localization of the RUSH constructs, cultures were incubated with 100 nM JF549 for 30–60 min at 37°C then rinsed twice prior to imaging. For concurrent ICC experiments, the JF dye JF549 was added to the secondary antibody mix and incubated at 25°C for 1 hr.

For the live-cell HaloTag pulse-chase labeling experiments, cultures were incubated with 1 nM JF549i for 6 days at 37°C, rinsed twice, and imaged. Incubation with JF549i for up to 8 days showed no detectable nonspecific uptake of this dye or crossing of the PM. JF549 was added to the coverslip at a final concentration of 100 nM during imaging.

## Live-cell imaging

Prior to imaging, RUSH reporter proteins were labeled with JF549 HTL (Janelia Farms) and, for rat neurons, anti-pan-neurofascin antibody (UC Davis/NIH NeuroMab Facility, A12/18; RRID:AB_2877334) for 60 min. Coverslips were rinsed twice with warmed PBS and returned to conditioned NBM. Coverslips were incubated with IgG2α Alexa Fluor 647 secondary (Thermo Fisher Scientific, A-21241; RRID:AB_2535810) for 15–30 min to label the anti-pan-neurofascin primary antibody. Coverslips were rinsed twice with warmed PBS and imaged in standard extracellular fluid (ECF) imaging solution (140 mM NaCl, 5 mM KCl, 2 mM $CaCl_2$, 2 mM $MgCl_2$, 5.5 mM glucose, 20 mM HEPES [pH 7.3] in PBS) at 37°C. The reporter proteins were released from the ER-localized streptavidin "hook" with the addition of 40 µM biotin (Sigma-Aldrich, B4639-100MG) to the coverslip. Biotin was diluted in 200 µl of ECF imaging solution and added to 800 µl of media in the imaging chamber for a final concentration of 40 µM. Videos were acquired ~20–30 min after biotin addition at 1 frame per second with a Zeiss 880 Airyscan LSM microscope and 63× objective using Fast Airyscan mode. All images were processed with automatic Airyscan deconvolution settings. Temperature, $CO_2$, and humidity were controlled using an Oko-lab incubation system.

## Kymograph generation and analysis

Kymographs (20 µm) were generated from 60 s RUSH movies from the soma-out direction using ZEN blue software 3.0 (ZEISS; Oberkochen, Germany), and were analyzed manually in Fiji (*Schindelin et al., 2012*). Directionality, as well as distance and time parameters, was recorded for each vesicle movement identified in the kymographs. For all figures, kymograph lines with a negative slope represent anterograde transport, and those with a positive slope indicate retrograde transport. The researcher was blinded to the kymographs analyzed in *Figure 4*.

## Immunocytochemistry

Dissociated cultures were fixed with 4% paraformaldehyde, permeabilized with 0.2% saponin, blocked (0.04% saponin, 10% goat serum, and 1% BSA in PBS), and then immunostained at 4°C (0.1% BSA and 0.04% saponin in PBS) overnight. The following morning, coverslips were rinsed three times for 5 min intervals with PBS and incubated with secondary antibodies (0.1% BSA and.04% saponin in PBS) for 1 hr. Then coverslips were rinsed three times for 5 min intervals with PBS and mounted on microscope slides (Thermo Fisher Scientific, 22-178277) using ProLong Glass Antifade Mountant (Thermo Fisher Scientific, P36980) or ProLong Glass Antifade with Mountant with NucBlue Stain (Thermo Fisher Scientific, P36981).

## Colocalization analysis

Pearson's correlation coefficients and Mander's coefficients were calculated using Fiji for ImageJ and Just Another Colocalization Plugin (JaCoP) (*Schindelin et al., 2012*; *Bolte and Cordelières, 2006*). Briefly, neurons were cultured, fixed and stained as described in the ICC methods, and imaged. Colocalization coefficients were measured for single optical sections.

## Expression level analysis

Hippocampal cultures were transduced with varying amounts of HaloTagged reporter virus on 9 DIV. On 14–16 DIV, cells were fixed and incubated with the JF549 HTL to label the reporter. Untransduced

control cultures, and cultures expressing the SYT1 and SYB2 reporters, were stained with α-SYT1 and α-SYB2 antibodies; as an internal control, all samples were also stained with an α-SYP antibody. The ICC protocol is detailed above, and the antibodies are detailed in the table shown under the Antibodies section, below.

With this paradigm, the SYT1 and SYB2 antibodies detect both the native and the tagged proteins such that the fluorescence difference at each synapse, indicating differences in protein quantity, can be compared. To this end, within the same field of view, ROIs of consistent size were used to measure the fluorescence intensity of the α-SYT1 or α-SYB2 signals at synapses with and without expression of the reporter protein (visualized with the JF549 HTL). These values were normalized to the fluorescence intensity of the ROIs in the α-SYP channel to control for variation in synapse size and intensity. Average relative expression levels were calculated by dividing the normalized fluorescence intensity of the α-SYT1 or α-SYB2 channel in synapses that had both the endogenous and tagged proteins, by the values obtained from synapses expressing only the native proteins.

## Protein immunoblots

Neuronal cell lysates were collected from dissociated neuronal cultures with 150 µl lysis buffer 2% SDS, 1% Triton X-100, and 10 mM EDTA in PBS, plus (1:200) 250 mM PMSF, and (1:500) 1 mg/ml aprotinin, leupeptin, and pepstatin A protease inhibitors. Samples were boiled at 100°C for 5 min after the addition of 50 µl of sample buffer (DTT, glycerol, and bromophenol blue) and 20 µl of lysates were run on 13.5% acrylamide gels with 10% 2,2,2,-trichloroethanol (TCE) (Sigma-Aldrich; T54801-100G). After protein separation by SDS-PAGE, the TCE was activated by UV light (300 nm) and the cross-linked proteins were imaged with a ChemiDoc MP Imaging System (Bio-Rad Laboratories) as a loading control (*Ladner et al., 2004*). SDS-PAGE gels were transferred to a PVDF membrane (Immobilon-FL; EMD Millipore) for 30 min per gel at a constant 240 mA, then blocked with 5% nonfat milk protein in Tris-buffered saline plus 1% Tween 20 (TBST) for 30 min. PVDF membranes were incubated in primary antibody, diluted in 1% milk in TBST, overnight at 4°C. The next day the membrane was rinsed and incubated with a secondary antibody, also diluted in 1% milk in TBST, for 1 hr, then washed three times for a total of 15 min. All washes were done with TBST. Immunoblots were imaged using Luminata Forte Western HRP substrate (EMD Millipore; ELLUF0100) and a ChemiDoc MP Imaging System (Bio-Rad Laboratories). Bands were analyzed by densitometry and contrast was linearly adjusted for publication using Fiji (*Schindelin et al., 2012*).

## Electrophysiological recordings

Whole-cell voltage-clamp recordings of cultured mouse hippocampal neurons (14–16 DIV) were performed at room temperature in ECF along with an internal pipette solution containing (in mM): 130 potassium gluconate, 10 HEPES, pH 7.4, 1 EGTA, 2 ATP, 0.3 GTP, 5 phosphocreatine. Recordings were performed using a MultiClamp 700B amplifier and Digidata 1550B digitizer (Molecular Devices, San Jose, CA) under the control of Clampex 10 software (Molecular Devices). AMPAR-mediated miniature excitatory post-synaptic (mEPSC) currents were pharmacologically isolated by including gabazine (50 µM) (Tocris Bioscience, Bristol, UK), D-AP5 (50 µM) (Tocris), and tetrodotoxin (1 µM) (Tocris) in the bath solution. QX 314 chloride (5mM) (Tocris) was included in the pipette solutions for all recordings. Neurons were held at –70 mV in all experiments without correction for liquid junction potentials. Recordings were discarded if series resistance rose above 15 MΩ; 180 s of data were recorded for each neuron. mEPSCs were quantified for each recording using a template-matching algorithm in Clampfit (Molecular Devices).

## Statistics

Exact values from experiments and analyses, including the number of data points (n) and number of trials, are included in the figures or are listed in the figure legends. Analyses were performed using GraphPad Prism 9.20 (GraphPad Software Inc). Normality was assessed by histograms of data and QQ plots; if normal, parametric statistical methods were used, if not, nonparametric methods were used for analysis. For all figures, $*p \leq 0.05$, $**p \leq 0.01$, $***p \leq 0.001$, $****p \leq 0.0001$; ns indicates $p>0.05$.

## Antibodies

### Primary antibodies

| Antibody | Source | Identifier | Concentration |
| --- | --- | --- | --- |
| Anti-SYT1 (48) | Developmental Studies Hybridoma Bank | Cat# mAB 48 (asv 48) RRID:AB_2199314 | ICC (1:500) IB (1:500) |
| Anti-SYB2/VAMP2 (69.1) | Synaptic Systems | Cat# 104211 RRID:AB_2619758 | ICC (1:500) IB (1:1K) |
| Anti-GM130 | BD Transduction Laboratories | Cat# 610822 RRID:AB_398142 | ICC (1:500) |
| Anti-pan-neurofascin (extracellular) antibody (A112/18) | NeuroMab | Cat# 75–172 RRID:AB_2282826 | ICC (1:1K) |
| Anti-SYP | Cedarlane Labs | Cat# 101004(SY) RRID:AB_1210382 | ICC (1:500) IB (1:1K) |
| Anti-MAP2 | Sigma-Aldrich | Cat# AB5543 RRID:AB_571049 | ICC (1:250) |

### Secondary antibodies

| Antibody | Source | Identifier | Concentration |
| --- | --- | --- | --- |
| Goat anti-Mouse IgG2α-Alexa Fluor 647 | Thermo Fisher Scientific | Cat# A21241 RRID:AB_2535810 | ICC (1:500) |
| Goat anti-Mouse IgG2β-Alexa Fluor 647 | Thermo Fisher Scientific | Cat# A21242 RRID:AB_2535810 | ICC (1:500) |
| Goat anti-Guinea Pig IgG-Alexa Fluor 647 | Thermo Fisher Scientific | Cat# A21450 RRID:AB_2735091 | ICC (1:500) |
| Goat anti-Mouse IgG2β-Alexa Fluor 488 | Thermo Fisher Scientific | Cat# A21141 RRID:AB_2535778 | ICC (1:500) |
| Goat anti-Mouse IgG1-Alexa Fluor 488 | Thermo Fisher Scientific | Cat# A21121 RRID:AB_2535764 | ICC (1:500) |
| Goat anti-Chicken IgG-Alexa Fluor 405 | Abcam | Cat# ab175675 RRID:AB_2810980 | ICC (1:500) |
| Goat anti-Mouse IgG-HRP | Bio-Rad Laboratories | Cat# 1706516 RRID:AB_11125547 | IB (1:10K) |
| Goat anti-Mouse IgG2β-HRP | Bio-Rad Laboratories | Cat# M32407 RRID:AB_2536647 | IB (1:10K) |
| Goat anti-Guinea Pig IgG-HRP | Abcam | Cat# ab6908 RRID:AB_955425 | IB (1:10K) |

## Resource availability

### Lead contact

Further information and requests for resources and reagents should be directed to and will be fulfilled by the Lead Contact, Dr Edwin Chapman (chapman@wisc.edu).

### Materials availability

All unique/stable reagents generated in this study are available from the Lead Contact with a completed Materials Transfer Agreement.

## Acknowledgements

We would like to thank the members of the Chapman lab, and K Drerup, for valuable discussion and feedback regarding this manuscript. We thank M Bradberry for the SYP-GFP fusion construct. This study was supported by grants from the NIH (MH061876 and NS097362 to ERC). JDV was supported

by a postdoctoral fellowship from the NIH (NS098604) and the Warren Alpert Distinguished Scholars Fellowship. ERC is an Investigator of the Howard Hughes Medical Institute. This article is subject to HHMI's Open Access to Publications policy. HHMI lab heads have previously granted a nonexclusive CC BY 4.0 license to the public and a sublicensable license to HHMI in their research articles. Pursuant to those licenses, the author-accepted manuscript of this article can be made freely available under a CC BY 4.0 license immediately upon publication.

## Additional information

### Funding

| Funder | Grant reference number | Author |
|--------|------------------------|--------|
| National Institutes of Health | MH061876 | Edwin R Chapman |
| National Institutes of Health | NS097362 | Edwin R Chapman |
| Howard Hughes Medical Institute | Investigator | Edwin R Chapman |
| National Institutes of Health | NS098604 | Jason D Vevea |
| Warren Alpert Foundation | Distinguished Scholars Fellowship | Jason D Vevea |

The funders had no role in study design, data collection and interpretation, or the decision to submit the work for publication.

### Author contributions

Emma T Watson, Conceptualization, Data curation, Formal analysis, Supervision, Validation, Investigation, Visualization, Methodology, Writing – original draft, Project administration, Writing – review and editing; Michaela M Pauers, Michael J Seibert, Data curation, Formal analysis, Investigation, Writing – review and editing; Jason D Vevea, Conceptualization, Supervision, Methodology, Writing – review and editing; Edwin R Chapman, Conceptualization, Resources, Supervision, Funding acquisition, Visualization, Methodology, Writing – original draft, Project administration, Writing – review and editing

### Author ORCIDs

Emma T Watson http://orcid.org/0000-0002-5336-8170
Michaela M Pauers http://orcid.org/0000-0001-9760-7041
Michael J Seibert http://orcid.org/0000-0003-1619-1852
Jason D Vevea http://orcid.org/0000-0002-3068-973X
Edwin R Chapman http://orcid.org/0000-0001-9787-8140

### Ethics

Animal care and use in this study were conducted in accordance with the NIH Guide for the Care and Use of Laboratory Animals handbook. Protocols were reviewed and approved by the Animal Care and Use Committee (ACUC) at the University of Wisconsin-Madison (Laboratory, Animal Welfare Public Health Service Assurance Number: A33688-01).

### Decision letter and Author response

Decision letter https://doi.org/10.7554/eLife.82568.sa1
Author response https://doi.org/10.7554/eLife.82568.sa2

## Additional files

### Supplementary files
- MDAR checklist
- Source data 1. Source data for *Figure 5* and *Figure 3—figure supplement 1*.

## Data availability

Detailed summary statistics are included as supplementary tables for Figures 2, 3, 4, and 5. Raw immunoblot and gel images are attached as a supplementary zip file.

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
