## [Editor Report]

The authors explored a key question in nerve cell biology, i.e. how these highly polarized cells achieve the specific and differential distribution of proteins and organelles into their axonal and dendritic compartments – the study is an important step forward in this context. By using a very-low-level expression paradigm to express fluorescently tagged reporter proteins in neurons, a method to allow their triggered and 'synchronous' exit from the endoplasmic reticulum (RUSH), and live cell imaging, the authors describe a specific axonal trafficking pathway for the synaptic vesicle proteins Synaptotagmin-1 and Synaptobrevin-2. The corresponding evidence is compelling, and, furthermore, the authors' observation that even slightly excessive expression levels of the fluorescently tagged reporters occlude the specific axonal trafficking so that proteins distribute indiscriminately into axons and dendrites, explains why previous studies often failed to detect specific axonal trafficking of synaptic vesicle proteins. This study will be of interest to cell biologists and neuroscientists alike because (i) it provides a major advance in our understanding of nerve cell development and function, (ii) it demonstrates the usefulness of the RUSH approach in nerve cell biology, and (iii) it stresses the importance of tight control of reporter (over)expression, which is important in many other contexts.

---

## [Decision Letter]

**Decision letter after peer review:**

Thank you for submitting your article "Synaptic vesicle proteins are selectively delivered to axons in mammalian neurons" for consideration by *eLife*. Your article has been reviewed by 3 peer reviewers, one of whom is a member of our Board of Reviewing Editors, and the evaluation has been overseen by a Reviewing Editor and Richard Aldrich as the Senior Editor. The reviewers have opted to remain anonymous.

Essential revisions requiring new experiments or new analyses

1. Analogous to the data on Syt1, the authors should carefully document the co-localization of the Syb2 reporter with the ER prior to the RUSH trigger. These data seem to have been acquired but they are not documented.

2. Regarding the tagged TfR imaging experiments, the question arises as to whether the authors observed a correlation of the previously described pre-axonal exclusion zone (Farias 2015; doi.org/10.1016/j.celrep.2015.09.074). There might be a specialized compartment just before the axon initial segment where the entry of presynaptic cargo but not of postsynaptic cargo can be detected. Similarly, Song 2009 (doi.org/10.1016/j.cell.2009.01.016) described a 'filter' due to which presynaptic cargo moves more slowly at the start of the initial segment, where postsynaptic cargo seems to be rejected. Here, the question arises as to whether the authors detected any corresponding transport speed differences match with this previous observation, and whether upon presynaptic cargo overexpression the saturation of such a choke point due to slow migration into the axon might explain the spill-over into the dendritic compartment. The reviewers do not expect additional experiments along these lines, but further analyses of the already available data might be useful in addressing these issues.

3. Aspects of the experiments designed to show that the tested vesicle proteins are first transported to the plasma membrane and subsequently built into proper synaptic vesicles are problematic. (i) It is unclear whether TeNT was present throughout the experiment and why ratios were calculated by dividing by background fluorescence in the absence of the non-permeable dye. (ii) It is unclear whether the expression of the Syb2 reporter can override the TeNT treatment. (iii) It is unclear how complete the block of synaptic secretion is upon TeNT treatment. The conclusions are based on the notion of a complete block, but this is not shown. Further, the possibility of a VAMP2-independent vesicle pool is not discussed, and neither is the fact that spontaneous vesicle fusion persists – at least to some degree – in Syb2 KO neurons. The reviewers acknowledge that dealing with these issues is not trivial. A possibility would be to stimulate (e.g. with high-KCl or field stimulation) and use TeNT, tagged SYT1, and staining with the JD549i dye to demonstrate that there is no (detectable) stimulated enhancement of surface SYT1 that could then be labeled. Without additional evidence, the text on the corresponding part of the study needs to be toned down substantially to reflect the preliminary character of the corresponding data.

Essential revisions requiring changes to text or figures

4. Labs that routinely image single fluorophore-tagged proteins (XFPs) observe that only 70-80% of the expressed protein is properly folded with productive fluorescence. This is usually established by bleaching multimers of known stoichiometry and then estimating the probability of a dark subunit required to fit the data. The authors should discuss this issue and corresponding knowledge in the literature regarding the Halo-fusion proteins – with respect to the probability of a dark protein (folding, failed labeling, etc.). This is relevant for the estimated expression ratios relative to wild-type protein levels but would not change any conclusions of the present study.

5. It seems that there is no directional transport of TfR in dendrites at all (Figure S5). It is unclear how this can be explained, and there is a possibility that this issue challenges the conclusion that the authors' approach reliably detects dendritic transport. This issue should be clarified*discussed.

6. The fact that the omission of ER-targeted GFP improves the RUSH readout is potentially concerning. If the overloading of transport pathways is an issue, the problem arises as to what the RUSH approach itself does to these processes. This issue should be clarified/discussed.

7. It is unclear why the approach shown in Figure 4 (e.g. panel E) leads to less selective trafficking of the reporters. Further, the 'no-effect' of the C2AB deletion is borderline convincing. It seems that there is an effect – just not high enough n. This issue should be clarified*discussed.

8. Generally, the C2-domain deletion experiments and the PTM manipulations are merely interesting first steps towards mechanistic insights, exploring some requirements for trafficking without going into much depth. The text should be adapted to reflect the very preliminary character of the corresponding data and conclusions.

9. The cartoon in Figure 5B is somewhat confusing in view of the data obtained. It appears that a panel depicting the eventual redistribution of the labeled protein from the surface to internal pools, to illustrate why adding the JF549 later would not cause a large increase in fluorescence, would be more helpful. It would match the observed data rather than depict a scenario that was not observed. This issue should be clarified.

*Reviewer #1 (Recommendations for the authors):*

The authors explored a decade-old problem in nerve cell biology, i.e. the question of how these extremely polarized cells achieve the specific and differential distribution of proteins and organelles into their axonal and dendritic compartments.

The present study represents a major step forward in this context. By using a very-low-level expression paradigm to express fluorescently tagged reporter proteins in neurons, a method (RUSH – ER retention by using selective hooks) to allow their triggered and 'synchronous' exit from the endoplasmic reticulum, and subsequent live cell imaging, the authors describe a specific axonal trafficking pathway for the synaptic vesicle proteins Synaptotagmin-1 and Synaptobrevin-2. The corresponding evidence is compelling. Furthermore, the authors' observation that even slightly excessive expression levels of the fluorescently tagged reporters occlude the specific axonal trafficking so that proteins distribute indiscriminately into axons and dendrites, explains why previous studies failed to detect the specific axonal trafficking of synaptic vesicle proteins.

This study will be of interest to cell biologists and neuroscientists alike because it provides a major advance in our understanding of nerve cell development and function. Further, the paper demonstrates the usefulness of the RUSH approach in nerve cell biology, which will be of interest to many scientists in the field. Finally, the paper stresses the importance of tight control of reporter (over)expression, which is important in many other contexts.

Key Strengths: Powerful combination of reporters, RUSH, tight control of reporter expression, and stringent analysis.

Key Weakness: It cannot be excluded that the RUSH approach affects normal trafficking, e.g. by overburdening some trafficking pathways.

1. Analogous to the data on Syt1, the authors should carefully document the co-localization of the Syb2 reporter with the ER prior to the RUSH trigger.

2. It seems that there is no directional transport of TfR in dendrites at all (Figure S5). I am unsure how this can be explained, and I feel that this question challenges the conclusion that they can see dendritic transport with their approach. This issue should be clarified*discussed.

3. The fact that the omission of ER-targeted GFP improved the RUSH readout is confusing. If the overloading of transport pathways is an issue, one wonders what the RUSH approach itself does to these processes. This issue should be clarified*discussed.

4. It is unclear why the approach shown in Figure 4 (e.g. panel E) leads to less selective trafficking of the reporters. Further, the 'no-effect' of the C2AB deletion is borderline convincing. It seems that there is an effect – just not high enough n. This issue should be clarified*discussed.

5. I am a bit confused with regard to the TeNT expression. Shouldn't the expression of the Syb2 reporter override this? In view of this, I think, based on the experiments shown, that the conclusion that Syt1 and Syb2 are first trafficked to the plasma membrane and then incorporated into synaptic vesicles is still premature. Also, in this context, one would like to see how effective the TeNT expression was in stopping the synaptic vesicle cycle in the cells analyzed.

*Reviewer #2 (Recommendations for the authors):*

The efforts of the authors in addressing the concerns of this and the other reviewers are appreciated, and the revised manuscript is indeed improved by providing necessary controls and some new data (e.g. the new synaptobrevin data). Undoubtedly, the study addresses a relevant and still unresolved question of cellular neurobiology. Having said that, the authors still provide mainly sheer observational data, and the mechanistic level achieved remains rather shallow. I am not rigorously objecting publication of this study in *eLife*, but I am just still not fully convinced that their progress meets the necessary standards. I do see that using their methodology they can demonstrate preferential dendritic transport of TfR, but the absence of such behavior for their SV cargo does not a priori deliver sufficient evidence for an axon-selective selective delivery pathway, at least in my eyes.

Mechanistic depth in my eyes (and as suggested) could have been provided by genetically targeting SYT1 C2AB domains via single point mutation to address ca^2+^ or lipid dependence of the proposed SYT1 selective axonal trafficking in mammalian neurons. I am sorry to say that I am still of the opinion that deleting the full C2AB domains of SYT1 is a rather rough approach and that my concern concerning truncating about two-thirds of SYT1 remains. Such an extended deletion in my eyes might just very principally affect the proteins trafficking/targeting into vesicles and particularly the cell-biological identity of the carrier it is transported by. I am prepared for the argument that this is what they wanted to demonstrate, however, in the absence of any further molecular information and manipulation concerning the nature of their suggested selective delivery pathway operating in mammalian neuron axons this reviewer stays unconvinced concerning a truly selective character here. This is also for another argument: while C2AB domain deletion seems to increase absolute amounts of dendritic trafficking estimated by counting trafficking vesicles (Figure 4I), although, with very high variance, absolute axonal trafficking rates for my understanding were unchanged. Isn't this more arguing for a role of the C2AB domains in blocking dendritic trafficking rather than selective axonal trafficking? C2AB domain deletion seems to increase the traffic altogether (see argument above), correct? Figure 4J: SYT1-PMG mutants display predominant anterograde trafficking in dendrites, arguing for the role of SYT1 palmitoylation/glycosylation in regulating dendritic traffic. Is this what the authors imply here?

Figure S1E: It is appreciated that the authors provide a colocalization analysis of SYT1-reporter with KDEL and GM130. While 30 min after biotin addition the Pearson's coefficient for SYT1-reporter and GM130 increases, the Pearson's coefficient for SYT1-reporter and KDEL remains unchanged. How do the authors explain this observation?

---

## [Author Response]

Essential revisions requiring new experiments or new analyses1. Analogous to the data on Syt1, the authors should carefully document the co-localization of the Syb2 reporter with the ER prior to the RUSH trigger. These data seem to have been acquired but they are not documented.

We thank the reviewer for making this suggestion. In response to this concern, we now include these data in Figure 1 —figure supplement 2 in the revised manuscript.

2. Regarding the tagged TfR imaging experiments, the question arises as to whether the authors observed a correlation of the previously described pre-axonal exclusion zone (Farias 2015; doi.org/10.1016/j.celrep.2015.09.074). There might be a specialized compartment just before the axon initial segment where the entry of presynaptic cargo but not of postsynaptic cargo can be detected. Similarly, Song 2009 (doi.org/10.1016/j.cell.2009.01.016) described a 'filter' due to which presynaptic cargo moves more slowly at the start of the initial segment, where postsynaptic cargo seems to be rejected. Here, the question arises as to whether the authors detected any corresponding transport speed differences match with this previous observation, and whether upon presynaptic cargo overexpression the saturation of such a choke point due to slow migration into the axon might explain the spill-over into the dendritic compartment. The reviewers do not expect additional experiments along these lines, but further analyses of the already available data might be useful in addressing these issues.

This is an interesting question. We observed few vesicles in the pre-axonal exclusion zone (PAEZ), so our initial thought was that exclusion was not coming into play. More specifically, of the nine TfR transport vesicles observed in the proximal axon, seven passed through the PAEZ and became either stationary or moved in an anterograde direction within the AIS. Of the remaining two transport vesicles, one underwent retrograde movement within the PAEZ, while the other dithered back and forth within this region. These observations potentially indicate some exclusion of TfR cargo at a PAEZ but, again, this applied to only two vesicles out of a small pool of nine vesicles in total. To make this issue more transparent, we now provide a breakdown of these data in Figure 2 —figure supplement 4F of the revised manuscript.

Regarding filtering at the axon initial segment (AIS), we observed that SYB2-containing vesicles are transported more slowly in the proximal axon (which encompasses the AIS) as compared to the distal axon (which does not include the AIS). This was not the case for SYT1; there was no significant difference in the transport speed of this cargo in proximal or distal axons. This protein-specific difference suggests a role for the kinesin motors in proceeding through the AIS. These data are now included as Figure 2 —figure supplement 3 in the revised manuscript. The idea that a bottleneck at the AIS could cause axonal cargo to spill over into other compartments is interesting, but our thought is that with time, the bottleneck effect would likely dissipate, and polarization would be restored. This was not the case for SYT1 and SYB2; the overexpression artifact of dendritic localization of these synaptic vesicle (SV) proteins persisted.

3. Aspects of the experiments designed to show that the tested vesicle proteins are first transported to the plasma membrane and subsequently built into proper synaptic vesicles are problematic. (i) It is unclear whether TeNT was present throughout the experiment and why ratios were calculated by dividing by background fluorescence in the absence of the non-permeable dye. (ii) It is unclear whether the expression of the Syb2 reporter can override the TeNT treatment. (iii) It is unclear how complete the block of synaptic secretion is upon TeNT treatment. The conclusions are based on the notion of a complete block, but this is not shown. Further, the possibility of a VAMP2-independent vesicle pool is not discussed, and neither is the fact that spontaneous vesicle fusion persists – at least to some degree – in Syb2 KO neurons. The reviewers acknowledge that dealing with these issues is not trivial. A possibility would be to stimulate (e.g. with high-KCl or field stimulation) and use TeNT, tagged SYT1, and staining with the JD549i dye to demonstrate that there is no (detectable) stimulated enhancement of surface SYT1 that could then be labeled. Without additional evidence, the text on the corresponding part of the study needs to be toned down substantially to reflect the preliminary character of the corresponding data.

We apologize for this miscommunication. The tetanus toxin light chain virus (now abbreviated as TeTx-LC for clarity) was added to cultures at 5 DIV and we now clarify this in the Figure 5 legend, lines 636-637 of the revised manuscript. The western blot (Figure 5D), which shows the loss of SYB2 in cultures that were treated with the virus, was conducted at 15 DIV, which was the day the experiment concluded. So, TeTx-LC was expressed and active throughout the experiment. The change in fluorescence ratios were calculated by dividing the final fluorescence by the initial fluorescence. This ratio was only calculated at synapses, not the entire field of view, to reduce the influence of the unbound permeable dye. The control condition, where we did not add nonpermeant dye, was divided by the background fluorescence in synapses to rule out contributions from autofluorescence and to remain consistent with the ratios calculated for the experimental condition. The background, in the control and test conditions, was reproducible and was a non-zero value, enabling us to calculate ratios. Also, this background signal was present in both conditions, so our analysis ensures these two conditions can be rigorously compared. The approach also helped to control for variation between trials. These issues are now clarified in the revised manuscript:

“The signal from labeling with the non-permeant ligand was referred to as F_initial_, where the unlabeled control coverslips still yielded a small background signal, producing a reproducible non-zero value that allowed us to calculate ratios.” – Lines 727-729

Regarding the reviewer’s second point, it is unlikely the SYB2 reporter is overriding the action of TeTx-LC, as the SYB2 construct was sparsely transfected (~0.01% of cells were expressing the protein). So, while SV exocytosis in the transfected neurons might be restored to some extent (but please also see our further response, regarding this issue, below), we do not see how there could be any rescue of naturally occurring, action potential driven, recurrent activity throughout the coverslip. Moreover, our preliminary studies indicate it takes at least a few days for newly transported synaptic vesicle proteins to rescue release (when evoked release has been disrupted in, for example, SYT1 KO neurons, or neurons expressing TeTx-LC), further reducing the likelihood that transfected SYB2 would rescue mini frequency to a significant degree over the course of our 6-day experiments. The key point here is that the network formed by these cultured neurons is largely silent, as there is no evoked transmission (which we can infer from the ability of TeTx-LC to phenocopy the SYB2 KO, as shown in Figure 5 —figure supplement 2, which was added during revision), and few minis, in any of the cells in the network.

Further regarding the effect of TeTx-LC on synaptic activity, our lab has studied the effects of tetanus toxin for the past 26 years and, as was shown in the paper cited in our manuscript (Bao, et al., *Nature* 2019, PMID: 29420480) and in countless unpublished experiments, the addition of tetanus toxin eliminates nearly all evoked synaptic transmission and greatly diminishes spontaneous release. This experiment is routine in our lab, and we did a poor job of conveying how predictable and consistent the results are in the first version of the manuscript. Additionally, expressing TeTx-LC was the best option available to us, because it disrupts both evoked and spontaneous release. Knockout of Sec1/Munc-18 would have eliminated all SV release, however loss of this protein is lethal in neurons, so we proceeded with TeTx-LC.

We reiterate that we now include electrophysiology experiments in Figure 5 —figure supplement 2 of the revised manuscript to demonstrate the effect of the toxin in the experiments in our study. These experiments confirm that when SYB2 is undetectable by western blot, as shown in Figure 5D, spontaneous release events are low, approximately 0.19 Hz; this corresponds to a 95% reduction in mini frequency, which—again— phenocopies the SYB2 KO (Schoch et al., *Science* 2001, PMID: 11691998). Using the general assumption that each neuron makes ~10^3^ connections (Gulati, *Indian J Pharmacol* 2015, PMID: 26729946; Azevedo et al., *J Comp Neurol* 2009, PMID: 19226510), the release frequency per synapse would be 0.19 x 10^-3^ events/sec, which, over the course of 6 days, could result in 98 spontaneous fusion events per synapse. To put this in perspective, if each vesicle only fuses once, this value represents 23% of the SVs found in a nerve terminal, based on previous measurements of cultures in our lab (Liu et al., *J Neurosci* 2009, PMID: 19515907). This percentage is still a gross overestimate because newly delivered protein is unlikely to be incorporated into SVs in the absence of synaptic activity, as the SV cycle is largely quiescent in our experiments due to expression of TeTx-LC. Indeed, our preliminary experiments suggest that SYT1 does not make it on to fusogenic vesicles (that is, does not restore evoked fusion in SYT1 KO neurons) 24 hours after delivery to the presynapse. We estimate that it will take at least an additional 24 hours for functional incorporation of tagged proteins into SVs. So, in our labeling protocol, over 6 days, there is unlikely to be any functional rescue of SYT1 or SYB2 for at least 2 of those days; correcting for this yields 66 fusion events per synapse over the course of our experiments, which yields an upper limit of 16% of the SVs in a nerve terminal being labeled through activity, if each vesicle fuses only once. Not to speculate too much, but we believe that efficient, functional incorporation of SV proteins into SVs requires activity and SV recycling, which is a process that has been efficiently disrupted in our TeTx-LC experiments. We have added text to the revised manuscript to clarify this point:

“We cannot rule out that some tagged protein was delivered to an internal compartment, however, the all-or nothing labeling we observed with the non-permeant ligand gives no indication of an internal depot that was protected from the non-permeant dye. Additionally, it is unlikely that the residual minis that occur in the presence of TeTx-LC (5%) contribute significantly to labeling at the PM for two reasons. Namely, in the absence of activity, the SV cycle and SV reformation are stalled, so tagged protein is unlikely to be efficiently incorporated into newly-produced, fusogenic vesicles that are able to participate in spontaneous or evoked release. Second, if tagged protein was delivered to an internal compartment, only to be subsequently labeled at the PM, this would require a fast and efficient pathway for incorporation into fusion-competent vesicles that undergo spontaneous release. However, we have conducted preliminary experiments using RUSH to rescue synaptic neurotransmission in SYT1 KO neurons and found that incorporation of tagged protein into functional vesicles takes days. This is consistent with the model, alluded-to above, in which SV recycling drives incorporation of newly delivered proteins into SVs. While we cannot rule out that a small fraction of tagged protein could be labeled through the residual minis that occur in the presence of TeTx-LC, this is unlikely to contribute to a significant degree. Thus, we conclude the major pathway involves delivery of newly synthesized SV proteins to the PM.” – Lines 370-385

As the reviewer points out, there is still the potential for SYB2-independent release, likely mediated by tetanus-insensitive VAMP, also called VAMP7, which is expressed at low levels in axons after synaptogenesis (Coco et al., *J Neurosci* 1999, PMID: 10559389). We emphasize that, as our electrophysiology experiments demonstrate, the residual mini release rate is minimal (5% of the frequency observed in untreated neurons, a value that is consistent with the KO literature (Schoch et al., *Science* 2001, PMID: 11691998)) so our work suggests that the contribution of SYB2-independent release is minor, if there is any contribution at all. Please note that we mention tetanus-insensitive VAMP in lines 776-777 of the Discussion of our manuscript.

Essential revisions requiring changes to text or figures4. Labs that routinely image single fluorophore-tagged proteins (XFPs) observe that only 70-80% of the expressed protein is properly folded with productive fluorescence. This is usually established by bleaching multimers of known stoichiometry and then estimating the probability of a dark subunit required to fit the data. The authors should discuss this issue and corresponding knowledge in the literature regarding the Halo-fusion proteins – with respect to the probability of a dark protein (folding, failed labeling, etc.). This is relevant for the estimated expression ratios relative to wild-type protein levels but would not change any conclusions of the present study.

This is absolutely true, and we thank the reviewer for pointing this out. We are unaware of any studies that estimate this. However, we note that the co-translational folding of recombinant HaloTag is efficient, with approximately 91% of the protein folding, unfolding, and then 73% correctly refolding (Samelson et al., *Sci Adv.* 2018, PMID: 29854950). So, even under in vitro conditions, this protein has a strong intrinsic ability to fold, somewhat mitigating the concern that there is a large fraction of “dark protein” in eukaryotic cells. Regarding our study, we emphasize that the ratios of tagged to wild type protein were calculated using antibodies against the protein of interest, not the HaloTag, so the folding efficiency of HaloTag does directly not influence our calculations.

5. It seems that there is no directional transport of TfR in dendrites at all (Figure S5). It is unclear how this can be explained, and there is a possibility that this issue challenges the conclusion that the authors' approach reliably detects dendritic transport. This issue should be clarified*discussed.

We agree with the reviewer and thank them for allowing us to address this issue. When we initially began these experiments, we had difficulties with the TfR construct, and it did not behave as well in our system as the axonal reporters did; mainly, we found that the TfR construct leaked prior to adding biotin. Thus, there is a considerable amount of stationary TfR already in dendrites. Regardless, this experiment still makes the point that at steady state, TfR is dendritically polarized. If it would be preferable to the reviewer, we can remove this figure from the paper, though we believe it supports our findings and have left it in the current version of our study.

6. The fact that the omission of ER-targeted GFP improves the RUSH readout is potentially concerning. If the overloading of transport pathways is an issue, the problem arises as to what the RUSH approach itself does to these processes. This issue should be clarified/discussed.

This is an excellent point that we failed to explain clearly in the resubmission. In the early phases of this project, we routinely marked the ER with our GFP-KDEL construct to aid in our interpretation of the transport vesicles (i.e. whether they contained protein that was in transit to the Golgi). At the start of this project, we had concerns regarding ER stress. Specifically, the ER and reporter constructs were aggregating, and the transduced cells appeared somewhat unhealthy compared to the untransduced wild type cells. This is an issue our lab has experience with, as we have confirmed ER stress in our experiments before when necessary (Ruhl et al., *Nat Comm*. 2019, PMID: 31387992).

To minimize ER stress in our current study, we optimized our approach and titrated both our ER-targeted GFP construct and reporter proteins until the proteins no longer aggregated, the transduced cells had typical morphology, and biotin-triggered release occurred in our RUSH assay. Before this titration, we were unable to obtain efficient release from the ER/Golgi in our system. This indicated what we believe to have been significant ER stress (though our attempts to directly measure ER stress under these conditions were inconclusive). Once we used the ER marker to establish our findings in Figure 2, it was our preference to leave it out because, as the reviewer mentioned, there is always a concern of ER stress and we wanted to avoid expressing unnecessary proteins; in our view, the less “load” the better. Additionally, we found that leaving out the additional marker also made release via RUSH marginally more reliable (perhaps by mitigating low levels of ER stress, but this is somewhat speculative), so we chose to not include it in most of our experiments. In short, it is likely that low levels of ER stress were present when both the ER-targeted GFP and reporter proteins were expressed; however, once release was achieved, the transport results obtained with and without the addition of the ER marker were consistent between these two conditions, mitigating our concerns about the impact of ER stress on SV protein transport.

7. It is unclear why the approach shown in Figure 4 (e.g. panel E) leads to less selective trafficking of the reporters. Further, the 'no-effect' of the C2AB deletion is borderline convincing. It seems that there is an effect – just not high enough n. This issue should be clarified*discussed.

To recap, our experiments using the C2AB deletion mutant had the highest n of all three conditions, and our statistical analysis indicates that this mutant traffics differently from the wild type and SYT1-PGM mutant. The trend toward some degree of polarization of the C2AB deletion mutant, as pointed out by the referee, suggests that all polarized trafficking might not be encoded entirely within the C2AB domain. In short, the referee’s point is well taken, and we clarify this issue in the revised manuscript:

“In contrast, removing the C2-domains of SYT1 did not affect axonal transport, but rather increased transport into dendrites, thereby disrupting the polarized distribution of this protein. As mentioned under Results, these findings suggest the tandem C2-domains might act to suppress dendritic transport. This raises the possibility that these domains help to direct SYT1 to transport organelles that are specific to axons, while the truncated protein is targeted to vesicles that do not undergo polarized transport. Additionally, this deletion mutant was present throughout the plasmalemma of both axons and dendrites at steady state. Clearly, further study is needed; the deletion mutant impairs the polarized transport and distribution of SYT1, but there is still a trend toward axonal enrichment. Nevertheless, these initial findings point to a role for the C2-domains in polarized transport of this protein.” – Lines 458-467

We are confident in the observed changes in polarized transport because we saw such a large number of C2AB deletion mutant transport vesicles in dendrites (40 vesicles), as compared to WT (21 vesicles) and the other syt1 mutant (15 vesicles). When normalized to the number of cells observed, these values become 3.6 vesicles for the C2AB mutant and 2.1 and 1.9 vesicles for the wild type and PGM mutants, respectively. Apparently, the removal of C2AB allows the SYT1 remnant to enter dendrites, which is now a new area of study in our lab that we hope will start to reveal the underlying mechanism for the polarized trafficking of this protein.

8. Generally, the C2-domain deletion experiments and the PTM manipulations are merely interesting first steps towards mechanistic insights, exploring some requirements for trafficking without going into much depth. The text should be adapted to reflect the very preliminary character of the corresponding data and conclusions.

The referee makes a fair point. We included these data because other labs have published that these posttranslational modifications are crucial for the transport of SYT1 (Han et al., *Neuron* 2004, PMID: 14715137; Kang et al., *JBC* 2004, PMID: 15355980; Atiya-Nasagi *J. Cell Sci.* 2005, PMID: 15755799). We had assumed the palmitoylation and glycosylation sites were going to be important, and we were surprised that mutating these sites had a negligible effect on transport. This is also in sharp contrast to SYT7, another isoform of the synaptotagmin family, which our lab showed heavily relies on PTMs for its processing and transport (Vevea et al., *eLife* 2020, PMID: 34543184). With palmitoylation and glycosylation ruled out, we now start from square one to determine how SYT1 is sorted to axons. We did see some decrease in polarization without C2AB, so this is a place to start, but we agree that extensive study must now be done to reveal what motifs of SYT1 are responsible for its polarized transport, and to uncover other interactors in this transport pathway that selectively route SV proteins to axons. This is a topic we have immense interest in, and we will address it using chimeric proteins. As suggested by the referee, we clarify this matter and emphasize, in the revised manuscript, that we are only at the starting point concerning the underlying mechanism for the observed polarized transport.

“Clearly, further study is needed; the deletion mutant impairs the polarized transport and distribution of SYT1, but there is still a trend toward axonal enrichment.” – Lines 465-466

9. The cartoon in Figure 5B is somewhat confusing in view of the data obtained. It appears that a panel depicting the eventual redistribution of the labeled protein from the surface to internal pools, to illustrate why adding the JF549 later would not cause a large increase in fluorescence, would be more helpful. It would match the observed data rather than depict a scenario that was not observed. This issue should be clarified.

We appreciate the constructive feedback and have expanded the cartoon in Figure 5B to reflect the suggested changes.

Reviewer #1 (Recommendations for the authors):The authors explored a decade-old problem in nerve cell biology, i.e. the question of how these extremely polarized cells achieve the specific and differential distribution of proteins and organelles into their axonal and dendritic compartments.The present study represents a major step forward in this context. By using a very-low-level expression paradigm to express fluorescently tagged reporter proteins in neurons, a method (RUSH – ER retention by using selective hooks) to allow their triggered and 'synchronous' exit from the endoplasmic reticulum, and subsequent live cell imaging, the authors describe a specific axonal trafficking pathway for the synaptic vesicle proteins Synaptotagmin-1 and Synaptobrevin-2. The corresponding evidence is compelling. Furthermore, the authors' observation that even slightly excessive expression levels of the fluorescently tagged reporters occlude the specific axonal trafficking so that proteins distribute indiscriminately into axons and dendrites, explains why previous studies failed to detect the specific axonal trafficking of synaptic vesicle proteins.This study will be of interest to cell biologists and neuroscientists alike because it provides a major advance in our understanding of nerve cell development and function. Further, the paper demonstrates the usefulness of the RUSH approach in nerve cell biology, which will be of interest to many scientists in the field. Finally, the paper stresses the importance of tight control of reporter (over)expression, which is important in many other contexts.Key Strengths: Powerful combination of reporters, RUSH, tight control of reporter expression, and stringent analysis.Key Weakness: It cannot be excluded that the RUSH approach affects normal trafficking, e.g. by overburdening some trafficking pathways.

We agree. This was something we were concerned about from the start and took all possible care to avoid— specifically by using the lowest amount of virus possible to minimize protein expression. Indeed, as was shown in Figure 3 of the manuscript, the protein levels we used were indistinguishable from wild type. Additionally, the cultures were monitored for signs of poor health, as indicated by aggregation of the tagged proteins in the ER, and the reporter constructs were also added later in development (9 DIV) to minimize the amount of protein that would be produced, and thus retained, in the ER prior to its release during experimentation. Furthermore, we only selected the dimmest neurons, indicating lower expression levels, and neurons with typical morphology as compared to their untransduced counterparts. Our efforts to use minimal expression levels are included in lines 137-142 of the manuscript.

“It is known that overexpression can cause SV proteins to mislocalize to other compartments, especially the PM (Pennuto 2003). To mitigate this confound, the viruses used to express SYT1 and SYB2 were carefully titrated to achieve a sparse transduction such that only a select few neurons were expressing minimal levels of the tagged protein. To further ensure low levels of expression, cells that had lower than average fluorescence (as compared to other transduced cells on the coverslip) were selected for imaging.”

1. Analogous to the data on Syt1, the authors should carefully document the co-localization of the Syb2 reporter with the ER prior to the RUSH trigger.

We thank the reviewer for making this suggestion. In response to this concern, we now include these data in Figure 1 —figure supplement 2 in the revised manuscript.

2. It seems that there is no directional transport of TfR in dendrites at all (Figure S5). I am unsure how this can be explained, and I feel that this question challenges the conclusion that they can see dendritic transport with their approach. This issue should be clarified*discussed.

We have clarified this issue under “Essential Revisions”, point #5, in the section above, and direct the referee to that response.

3. The fact that the omission of ER-targeted GFP improved the RUSH readout is confusing. If the overloading of transport pathways is an issue, one wonders what the RUSH approach itself does to these processes. This issue should be clarified*discussed.

We thank the referee for this question. Please see our response to #6, under “Essential Revisions”, above.

4. It is unclear why the approach shown in Figure 4 (e.g. panel E) leads to less selective trafficking of the reporters. Further, the 'no-effect' of the C2AB deletion is borderline convincing. It seems that there is an effect – just not high enough n. This issue should be clarified*discussed.

We appreciate the opportunity to address this issue. Please see our response to #7, under “Essential Revisions”, above.

5. I am a bit confused with regard to the TeNT expression. Shouldn't the expression of the Syb2 reporter override this? In view of this, I think, based on the experiments shown, that the conclusion that Syt1 and Syb2 are first trafficked to the plasma membrane and then incorporated into synaptic vesicles is still premature. Also, in this context, one would like to see how effective the TeNT expression was in stopping the synaptic vesicle cycle in the cells analyzed.

We thank the referee for this question. Please see our response to #3, under “Essential Revisions”, above.

Reviewer #2 (Recommendations for the authors):The efforts of the authors in addressing the concerns of this and the other reviewers are appreciated, and the revised manuscript is indeed improved by providing necessary controls and some new data (e.g. the new synaptobrevin data). Undoubtedly, the study addresses a relevant and still unresolved question of cellular neurobiology. Having said that, the authors still provide mainly sheer observational data, and the mechanistic level achieved remains rather shallow. I am not rigorously objecting publication of this study in eLife, but I am just still not fully convinced that their progress meets the necessary standards. I do see that using their methodology they can demonstrate preferential dendritic transport of TfR, but the absence of such behavior for their SV cargo does not a priori deliver sufficient evidence for an axon-selective selective delivery pathway, at least in my eyes.Mechanistic depth in my eyes (and as suggested) could have been provided by genetically targeting SYT1 C2AB domains via single point mutation to address ca^2+^ or lipid dependence of the proposed SYT1 selective axonal trafficking in mammalian neurons. I am sorry to say that I am still of the opinion that deleting the full C2AB domains of SYT1 is a rather rough approach and that my concern concerning truncating about two-thirds of SYT1 remains. Such an extended deletion in my eyes might just very principally affect the proteins trafficking/targeting into vesicles and particularly the cell-biological identity of the carrier it is transported by. I am prepared for the argument that this is what they wanted to demonstrate, however, in the absence of any further molecular information and manipulation concerning the nature of their suggested selective delivery pathway operating in mammalian neuron axons this reviewer stays unconvinced concerning a truly selective character here. This is also for another argument: while C2AB domain deletion seems to increase absolute amounts of dendritic trafficking estimated by counting trafficking vesicles (Figure 4I), although, with very high variance, absolute axonal trafficking rates for my understanding were unchanged. Isn't this more arguing for a role of the C2AB domains in blocking dendritic trafficking rather than selective axonal trafficking? C2AB domain deletion seems to increase the traffic altogether (see argument above), correct? Figure 4J: SYT1-PMG mutants display predominant anterograde trafficking in dendrites, arguing for the role of SYT1 palmitoylation/glycosylation in regulating dendritic traffic. Is this what the authors imply here?

We note that mutant forms of SYT1 have been expressed in neurons since 2002, and point mutations in the Ca^2+^ and lipid binding loops, or the polybasic motifs of each C2-domain (e.g. Fernandez-Chacon 2002 *J Neurosci* 2002, PMID: 12351718; Stevens & Sullivan *Neuron* 2003, PMID: 12873386; Wu et al., *J Neurosci* 2022, PMID: 35701163), or in crucial residues (for release) at the “bottom” of the C2B domain (Xue et al., *Nat Struct Mol Biol* 2008, PMID: 18953334), as just a few examples, do not affect trafficking. The linker between the C2-domains has also been mutated extensively (Liu et al., *Nat Neuro* 2014, PMID: 24657966) and this did not alter vesicular targeting. In short, there are no specific residues to consider for mutagenesis and transport, and subtle deletions result in misfolding of the C2-domains. However, our lab has deleted each C2-domain individually and found much of the protein becomes stranded on the plasma membrane, with a fraction that is still targeted to SVs (Courtney et al., *Nat Comm* 2019, PMID: 31501440). Consequently, we started with this admittedly coarse experiment by deleting the entire C2AB domain. We were surprised that this deletion impaired the polarized transport of the protein; we had anticipated the N-terminal domain might encode all the information for axonal targeting. Hence, our finding sets the stage for chimeric analysis, which we are currently setting up.

Regarding the interpretation of the C2AB domain deletion data, we agree that are findings are somewhat coarse and could be viewed as somewhat preliminary. However, as alluded to in the previous paragraph, we also emphasize that the capability to affect polarized transport, at all, is an important first step and sets the stage for further structure-function experiments. As the referee mentioned, the total amount of axonal transport of the C2AB deletion mutant is relatively unchanged in comparison to the wild type, but this construct was now transported into dendrites. This suggests that the C2-domains may act by negatively regulating the incorporation of SYT1 into subsets of transport vesicles destined for dendrites, because, as the referee pointed out, we observed an increase in overall transport when these domains are removed. We have revised the manuscript to reflect these points, noting the increase in dendritic transport concurrent with normal levels of axonal transport.

“Interestingly, the C2AB deletion mutant resulted in increased dendritic transport as compared to the WT protein, while axonal transport remained unchanged, indicating these domains might play a role in targeting SYT1 to different subsets of transport vesicles with distinct destinations.” Lines 303-306

As for the dendritic transport of the SYT-PGM, we are hesitant to draw conclusions based on the low number of transport vesicles observed in dendrites. This point is clarified/addressed in lines 313-316 of the manuscript.

“Interestingly, SYT1-PGM overwhelmingly moved in an anterograde direction in dendrites under the nonequilibrium conditions of these experiments. However, the total number of transport vesicles carrying SYT1 and SYT1-PGM in dendrites was relatively low, so the observed differences should be interpreted with caution.”

Figure S1E: It is appreciated that the authors provide a colocalization analysis of SYT1-reporter with KDEL and GM130. While 30 min after biotin addition the Pearson's coefficient for SYT1-reporter and GM130 increases, the Pearson's coefficient for SYT1-reporter and KDEL remains unchanged. How do the authors explain this observation?

We thank the referee for the opportunity to clarify this point. Based on our observations, the ER is large enough that, in 30 minutes, not enough of the total SYT1 reporter has left the ER for a substantial reduction in the Pearson’s coefficient.